# Traffic impact modelling in SURFEX-TEB V9.0 model for improved road surface temperature prediction

Gabriel Colas[1], Valéry Masson[1], François Bouttier[1], and Ludovic Bouilloud[2]

[1]CNRM, Université de Toulouse,Météo-France, CNRS, Toulouse, France
[2]Météo-France, Toulouse, France

**Correspondence:** Gabriel Colas (gabriel.colas@meteo.fr)

**Abstract.** The impact of road traffic on local climate has often been overlooked, being modelled as an aggregated sensible heat flux released into the atmosphere, although it has multiple effects including turbulence, heat from energy inefficiencies of vehicles, tyre friction, snow compaction, and shadowing. These effects can impact road surface conditions and exacerbate the phenomenon of Urban Heat Island (UHI). This study aims to improve the representation of traffic impacts in the Town Energy
Balance (TEB) V9.0 urban climate model. Particular attention has been paid to preserve physical consistency among the parameterisations of tyre friction, turbulence, energy inefficiencies, and radiation impacts of the road traffic within the model. In addition, a method has been developed to model the average engine efficiency of the entire automobile fleet with internal combustion engines (ICEs) using the Worldwide Harmonized Light vehicles Test Cycles (WLTC). The new parameterisations are evaluated using observations from two road weather stations in southern Finland, Nupuri and Palojärvi, which are characterised
by clear commuting patterns. To evaluate the new traffic parameterisation, road surface temperature (RST) differences between the two road carriageways are used to isolate the traffic-induced effects from the natural factors. The results show that the new parameterisation is able to simulate the traffic-induced impacts on road surface temperatures. In addition, wind-induced impact and rolling friction have been shown to drive traffic effects on RST. Taking explicitly into account the traffic impacts might be better suited to simulate their actual impacts on the local scale.

*Copyright statement.*

## 1 Introduction

Road traffic has increased from 4.5 trillion passenger kilometres travelled in 1995 to 6 trillion in 2019, and from 2.4 billion tonne-km travelled for freight in 1995 to 3.3 billion in European Union in 2019 (EEA, 2024). The transport sector is a massive source of greenhouse gases, with the land transport sector being the fourth largest contributor, with long-term effects on global
climate (IPCC, 2022). In addition to the long-term impacts on global climate, the cumulative effects of local dense road traffic significantly influence local climate and air quality. Road traffic is an important source of heat, pollution, turbulence, and friction with the surface, which impact the local energy balance. Cities that concentrate both a large population and high

traffic are particularly impacted. At rush hour, some road segments on the Paris ring road can reach up to 220 0000 vehicles a day (Amato et al., 2016). No studies have attempted to assess or simulate the complete set of traffic impacts on local climate,
which reveals a gap in existing weather and climate modelling tools.

A significant amount of the primary energy source of the motor vehicle transformed into mechanical power is lost and released as sensible and latent heat in the atmosphere. For vehicles equipped with internal combustion engines, that is, approximately 98% of the total automobile fleet in Europe in 2023 (Eurostat, 2023), more than 75% of the fuel combustion energy is lost (Johnson and Joshi, 2018). It is released as heat in the urban canopy along with house heating, air conditioning
systems (de Munck et al., 2013) and energy loss from industries. In cities, traffic can be an important contributor to the total anthropogenic heat released in the air. However, by adopting an electric-based vehicle fleet, the total anthropogenic heat released decreases proportionally (Chen and Yang, 2022). The relative contribution of traffic to anthropogenic heat is greater in summer than in winter (Bohnenstengel et al., 2013; Pigeon et al., 2007). However, the impact of traffic on the local climate is greater in winter, particularly at rush hours (Pigeon et al., 2007; Iamarino et al., 2011, Chen et al., 2021). Many methods are available
to estimate the heat released from the building sector (Bueno et al., 2012; Iamarino et al., 2011; Best and Grimmond,2016). On the contrary, there are few methods for the heat released by traffic. It can be modelled as an estimate aggregated with the other sources of anthropogenic heat (Varquez et al.,2021; Flanner,2009) or modelled separately (Pigeon et al.,2007; Ward et al., 2022; Sailor and Lu,2004). Modelled separately, the heat released by traffic is estimated mainly from inventories approach, with the average fuel consumption over the entire transport sector (Kłysik, 1996), or from vehicle fuel consumption statistics
(Iamarino et al.,2011; Harrison et al.,1984; Pigeon et al.,2007) or from arbitrary estimates (Sailor and Lu, 2004). Most of the other impacts of traffic, such as turbulence, friction with the surface, and changes in local energy balance, are not taken into account.

A moving vehicle has direct effects on the surface energy balance as well as on local turbulence. The tyres oppose a rolling resistance to the direction due to their viscoelastic properties, leading to thermomechanical impacts. Tyres warm up mainly
due to the hysteresis effect (Lin and Hwang, 2004) and road warms up through friction and conduction (Kelly and Sharp, 2012; Logan, 2012). Thus, traffic can lead to increased road surface temperature (Fujimoto et al., 2008; Khalifa et al., 2016), snow compaction (Wahlin et al., 2014), and water splashing (Karsisto, 2024; Denby et al., 2013). In addition, a vehicle body is a moving obstacle immersed in the air with drag amount based on the structure of the flow in its wake (Ahmed, 1981). The local turbulence produced by vehicles directly influences the heat exchanges within the canopy with modified turbulent
heat exchange coefficients (Fujimoto et al., 2008; Khalifa et al., 2016; Denby et al.,2013). The vehicle body also has radiative impacts on the local environment, including decreased solar radiation received by the surface. The physics related to vehicle dynamics has been well studied, and many models have been developed to simulate their various components at different levels of complexity (Jazar, 2009), from analytical approximations (Pacejka, 2000) to extensive numerical calculations (Farroni et al., 2014). Despite the significant impact of traffic on the local scale, these tools have not yet been integrated into models that
simulate local surface climate conditions.

Two classes of Land Surface Models (LSMs) are suitable for including the traffic impacts introduced before on local climate conditions: urban climate models and road weather forecast models. Built to model the local conditions of artificial

environments such as cities or the road network, they are able to accurately simulate city-wide building energy consumptions (de Munck et al., 2013; Jin et al., 2021), urban heat island effect (UHI) (De Ridder et al., 2015; Lemonsu et al., 2015) or road surface conditions (Colas et al., 2024). Depending on the purpose of the model, several traffic processes have been included, but they often rely on oversimplification. Urban climate models, developed mainly to study the urban climate and to provide boundary conditions to atmospheric models, only include the heat released by traffic (Lipson et al., 2024). It is modelled as a simple parameterisation of diurnal heat released directly into the atmosphere as sensible or latent heat (Lipson et al., 2024). This source is often aggregated and inseparable from the other sources of anthropic heat. Khalifa et al. (2016) have made a first attempt to model the entire set of impacts of traffic heat on snow-free roads within the urban climate model Town Energy Balance (TEB). This attempt has been largely inspired by Fujimoto et al. (2008) with the RSV-SV road weather model. Traffic impact parametrisations are more widely used in road weather models, since traffic has a significant impact on road surface conditions. NORTRIP (Denby et al., 2013), RoadSurf (Karsisto, 2024), BJ-ROME (Meng, 2018) and METRo (Crevier and Delage, 2001) road weather models include simple parameterisations of the anthropic heat released by traffic as a diurnal sensible heat flux. The RoadSurf model from the Finnish Meteorological Institute also includes a parameterisation of traffic impact on the surface hydrology, through contact of the tyre with the surface: the amount of water or snow decreases exponentially with time through the spray and splash processes. This impact is also included in the Norwegian NORTRIP model with a formulation that depends on the speed and count of moving vehicles. In addition, the NORTRIP model includes the impact on turbulent exchange between the road surface and the air with modified exchange coefficients.

In this study, a new modelling strategy introduced in Sect. 2 is developed to take into account the traffic impacts in the LSMs. This approach is mainly developed to improve road surface conditions in winter, as traffic impacts in winter are larger than in summer. It is integrated into the SURFEX-TEB V9.0 urban climate model in order to improve the simulation of winter conditions (Colas et al., 2024). Parametrisations are developed for the heat released from engine inefficiencies and the surface-tyre interaction, impact of the vehicle body on the radiation budget, and impact on the turbulent heat exchange as presented in Sect. 3. New consistent formulations are derived from approximated analytical solutions that depend on vehicle density. The heterogeneity of the driving behaviours and vehicle models are also estimated and taken into account. This new parametrisation is evaluated at two locations chosen in Southern Finland, at road weather stations with atmospheric, surface, and vehicle counting observations. These experiments and the configurations of the model TEB are detailed in Sect. 4. The SURFEX-TEB V9.0 model with traffic impacts is then evaluated against road surface observations Sect. 5. Finally, Sect. 6 discusses the results of our modelling before the concluding remarks.

## 2 Modelling strategy

### 2.1 Including traffic impact parameterisations in TEB

In this article, TEB is improved with new parameterisations. The model will now take into account the heat released from engine inefficiencies, the impact of the vehicle body on the radiation budget and on the turbulent heat exchange, and the heat produced by the surface-tyre interactions, as shown in Fig. 1. The direct impacts of traffic on the water, ice, and snow cover are

not modelled. They do not fall in the scope of this study. As TEB is an horizontally averaged model, the model must integrate horizontally averaged traffic impacts. So, for example, at one model grid point, despite the significant heterogeneity of the road traffic, the wind induced by the entire vehicle fleet returns a single horizontal average value. Traffic intensity is included in the model through average values of traffic counts and converted to vehicles per second. Therefore, traffic counts change at each

atmospheric forcing time (each hour in this study).

Two energy budgets are parameterised: the internal energy budget and the vehicle body energy balance. The latter calculates the radiation impacts on the vehicle and, conversely, the impacts of the vehicle body on TEB energy balance. The former calculates the energy generated from the fuel combustion and transformed into mechanical energy or into heat. From the mechanical energy, a share is transferred to the road through the surface-tyre interaction. From the energy transformed into

heat, a first share is released as sensible and latent heat into the air and a second share warms the bottom vehicle body as shown in Fig. 1. Each share is estimated through the dynamics of a vehicle, modelled by a simple Newtonian mechanics equilibrium equation inspired by Bera (2019).

The internal energy budget and the vehicle body energy balance are coupled in a simple way: (1) The temperature of the bottom vehicle body surface is prescribed as in Fujimoto et al. (2008). It increases the infrared radiation emitted by the surface.

(2) The additional energy needed to increases the infrared radiation emission from the bottom of the vehicle body is extracted from the internal energy balance of a vehicle. Finally, the turbulent heat exchange coefficient between the road surface and the air is modified by the traffic-induced wind. A simple analytical formula is developed to calculate the wind induced by traffic inspired by the study of Eskridge and Hunt (1979).

If the goal is to the parameterised the impact of electric vehicles, it is possible through modifying the internal energy balance

parameterisation. Because of their much larger energy efficiencies (Weiss et al., 2020), it is possible to consider no energy loss from their internal energy balance. In practice, this means multiplying the heat released into the air and the infrared radiation from the warm surface of the bottom vehicle body by the proportion of non-electric vehicles.

The urban energy balance without traffic impacts is written as :

$$Q^* + Q_f = Q_h + Q_e + \Delta Q_s + Q_m + \Delta Q_a \tag{1}$$

In the urban area, $Q^*$ and $Q_f$ are the net radiative heat flux and the anthropic heat flux (without traffic), respectively. On the right-hand side of the equation, there are two turbulent fluxes $Q_h$ and $Q_e$ sensible and latent heat fluxes, respectively, and $\Delta Q_s$ the heat flux by conduction through the urban surfaces. $Q_m$ is the power exchanged by the melting and freezing of the water and finally $\Delta_Q a$ the horizontal advection, which is neglected in this study. All terms of the equation are expressed in watts per square metre.

Each traffic impact then modifies the energy balance within the canopy. First, three new source terms are added into this equation with the sensible $Q_{Hengine}$ and latent heat flux $Q_{Eengine}$ from the heat lost by the vehicle engine and the tyre-road friction heat flux $Q_r$. Secondly, the turbulent fluxes $Q_h$, $Q_e$ of the urban area are modified by the wind induced by the traffic $U_{traff}$. Indeed, traffic modifies the fluxes from the soil-atmosphere interaction. Finally, the solar and infrared net heat fluxes are modified by the vehicle body as depicted in Fig. 1 and the resulting energy from the vehicle balance is released as sensible

heat in the air. The other terms are not formally modified, but react according to the new energy exchanges driven by the new and modified terms. The updated urban energy balance, function of the different traffic impacts gives:

$$Q^*(traff) + Q_f + Q_{f\_traff} = Q_h(U_{traff}) + Q_e(U_{traff}) + \Delta Q_s + Q_m + \Delta Q_a + Q_{Hengine} + Q_{Eengine} + Q_r + Q_{veh} \quad (2)$$

with $Q_{f\_traff}$, the total source of energy coming from the vehicles distributed in the different right-hand terms of the equation.

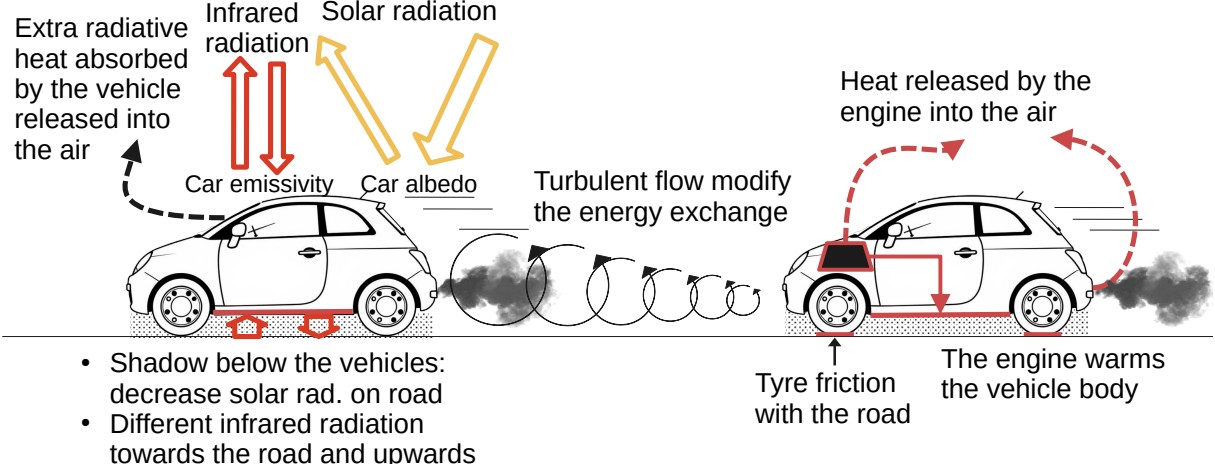

**Figure 1.** Simplified scheme of the traffic impacts parametererised included in the TEB model with on the left the processes related to the impact of the vehicle body and on the right the processes related to the internal operation of a vehicle

## 2.2 Traffic heterogeneity modelling within the parameterisations

The impact of traffic on local climate must be estimated considering the large heterogeneities of vehicle type, size, internal parts, engines, and driver behaviour. On one hand, traffic is a collection of vehicles with various characteristics: size, shape, internal parts, and engine type. From the distribution of all existing characteristics of the vehicles, a finite set of variables and parameters is defined. The inherent physical characteristics of a vehicle can be defined by its weight $m$ (kg), length $l$ (m), height $h$ (m), cross-section $A$ (m$^2$), energy power efficiency $\eta$, and drag coefficient $C_d$. They are assumed to be the characteristics of

a vehicle that influence the most the traffic impacts on local climate. On the other hand, at each location, traffic is a collection of driving behaviours, which can range from economical (with slow accelerations) to more aggressive. This variability in behaviour is modelled by the speed $v$ and the acceleration $a$ of a vehicle.

The vehicle characteristics are assumed to be independent from the driver behaviour. This assumption is reasonable, as most vehicles can reach the maximum speed allowed in all countries. However, energy power efficiency cannot be assumed to be

independent from the driver behaviour: engine efficiency $\eta_e$ depends directly on speed, acceleration, gear choice and engine type. Thus, the following assumptions are made: (1) The distribution of vehicle body characteristics is specified through its average values $(\overline{m}, \overline{l}, \overline{h}, \overline{A}, \overline{C_d})$ as shown in Fig. 2. They are independent from the other variables and represent a vehicle that has the average characteristics of the overall automobile fleet. (2) An average vehicular speed $\overline{v}$ is measured at a specific

location, but the distribution of the speed and acceleration of the vehicle fleet at this location is unknown. So, the relationships
between $\bar{v}$ and the distribution the of speed and acceleration are inferred from other data sources. They are then inserted into the
traffic impact equations as explained in the next paragraph and in Appendix B to take into account the impact of the collection
of driver behaviour. (3) Because instantaneous engine efficiency depends on each vehicle speed, and the vehicle fleet has a
distribution of speed, an average engine efficiency $\overline{\eta_e}(\bar{v})$ of the total automobile fleet depending on the average speed $\bar{v}$ is
computed. It is estimated from other input data sources, as shown in Fig. 2.

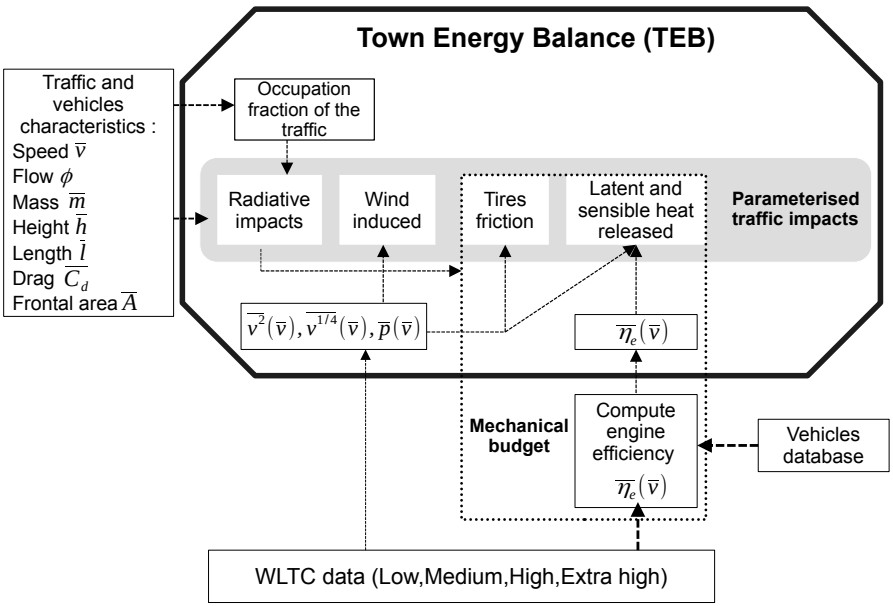

**Figure 2.** Protocol of the study to integrate the traffic impacts in the model TEB with the use of the World Light vehicles Test Cycles (WLTC)
data

The collection of driver behaviour and its impact on traffic impact is estimated using the Worldwide Harmonised Light
vehicles Test Cycles (WLTC). These cycles have been designed to provide a common and reliable measure of the energy
consumption of all vehicles sold in the European Union. Since 2019, for every vehicle sold, car manufacturers have the legal
obligation to provide detailed vehicle energy consumptions measured with the WLTC. For passenger cars, researchers designed
four subcycles, $s = \{1, 2, 3, 4\}$, of low-speed, medium-speed, high-speed, extra-high speed representative of different road
types. In this study, these cycles are considered to be associated with urban areas, suburban areas, rural areas, and highways,
respectively. Each WLTP cycle has been built from a speed and acceleration data sample of the world's driving habits (Tutuianu
et al., 2015).

## 3  Methods

In TEB, the area occupied by each component (buildings, gardens, snow-cover) is defined by an occupation fraction. The same strategy is used for the traffic. Vehicles cover a fraction of the road area $f_{traff}$. In agreement with the horizontally average modelling in TEB, $f_{traff}$ represents the proportion of a segment perpendicular to the road covered by traffic. Vehicles are considered to drive in the snow-free part of the road only. Thus, when snow covers the road, $f_{traff}$ is weighted by the snow fraction $f_{sn}$ calculated as in Colas et al. (2024). With the traffic flow $\Phi$ (vehicles s$^{-1}$) and average traffic speed $\overline{v}$ (m s$^{-1}$) punctual measures $f_{traff}$ is defined as:

$$f_{traff} = (1 - f_{sn}) \frac{\overline{w}}{w_{rd}} (\frac{\overline{l}}{\overline{v}} \Phi) \qquad (3)$$

with $\overline{w}/w_{rd}$ the lateral occupation of the road traffic calculated with $w_{rd}$ the road width and $\overline{w}$ the average vehicle width and $\overline{l}$ the average vehicle length.

### 3.1  Traffic impact on the radiative budget

Each vehicle is modelled as a flat 2-side surface, each with a different surface temperature. The upper vehicle body temperature is modelled equal to the air temperature $T_{can}$ (K) of the lower air layer of TEB. The bottom vehicle body temperature warmed by the vehicle engine is modelled as in Fujimoto et al. (2008) equal to $T_{can} + 22.5 K$. In addition, vehicles are modelled as a new component at ground level which shade the road, contributes to infrared emissions and radiation inter-reflections within the canyon calculated as in Lemonsu et al. (2012). Immersed in the urban canyon of TEB, the traffic takes part in the radiative exchange with the other TEB components: road, walls, windows, ground-based vegetation, and tree canopy. The footprint of traffic impact on energy exchanges is assumed to be strictly equal to the total area occupied by the vehicle fleet. Thus, each vehicle impact on the TEB energy exchanges is aggregated with the traffic occupation fraction $f_{traff}$, which represents the entire vehicle fleet.

The vehicle body energy budget composed of the shortwave and longwave radiation is solved. In addition, each vehicle is modelled without explicit thermal capacity: it means that the extra energy from the radiative budget absorbed by the vehicle body is transferred directly as sensible heat directly into the urban canyon air bottom layer. It is written as follows:

$$S^*_{veh} + LW^*_{veh} = Q_{veh} \qquad (4)$$

where $S^*_{veh}$ (W m$^{-2}$) is the shortwave radiation absorbed by a vehicle, $LW^*_{veh}$ (W m$^{-2}$) the longwave radiation absorbed by a vehicle and $Q_{veh}$ (W m$^{-2}$) the residual energy drive by the radiative balance excess that is transferred as sensible heat directly into the urban canyon air bottom layer.

The solar radiation received by a vehicle is modelled as the sum of three terms: the direct $S^{\Downarrow}_r$ and diffuse solar radiation $S^{\downarrow}_r$ received at the ground level and the infinite solar reflection within the urban canyon. A share of the solar radiation received by a vehicle is then reflected by the vehicle albedo $\alpha_{veh}$. In addition, the reflection within the urban canyon are modified by the

new aggregated surface albedo $\overline{\alpha_g}$ written :

$$\overline{\alpha_g} = (1 - f_{sn})(1 - f_{traff})\alpha_{rd} + f_{traff}\alpha_{veh} + f_{sn}\alpha_{sn} \tag{5}$$

With $f_{sn}$ the fraction of the snow occupation on the road, $\alpha_{sn}$ the albedo of snow, $\alpha_{rd}$ the albedo of the snow-free road surface and $f_{traff}$ the traffic occupation fractions defined Eq. (3).

The longwave exchanges are computed following a linear approximation of the Stefan–Boltzmann law as in Lemonsu et al. (2012). As for the solar radiation, TEB solves the energy budget from the infrared radiation for each component. The vehicle longwave exchange are calculated with the walls, the windows, the sky, the road and summed over to give the total
longwave radiation absorbed by a vehicle. Each component in TEB is also impacted by the vehicle at ground level: for each component in TEB, the additional exchange with vehicle body is added to the longwave exchanges.

## 3.2 Vehicle's internal heat and tyres friction

Each vehicle is an autonomous system with its own internal behaviour and physical response. It affects the physical variables of the street while passing with heat relased in the atmosphere and with heat transferred to the road surface from tyre friction. The
heat relased in the atmosphere is lost from the combustion of fuel in an internal combustion engine (ICE) due to mechanical inefficiencies $\eta_m$ from all the mechanical frictions and rotating parts of the vehicle and the thermodynamic inefficiencies of its engine $\eta_e(n, M)$. The product of both $\eta_m$ and $\eta_e(n, M)$ gives the total instantaneous vehicle efficiency coefficient $\eta$. The thermodynamical inefficiencies of its engine depending on the engine rotation $n$ (rpm) and torque $M$ ($\mathrm{N\,m^{-1}}$) are a key variable for representing the energy lost by vehicles and for tracking changes in energy performance over the years (Johnson and Joshi,
2018). The maximum engine efficiency $\eta_{emax}$ can be reached with a specific engine rotation speed and torque, but most of the time the engine efficiency of the vehicle is lower. A vehicle power-efficiency can either be directly measured (Newman et al., 2015; Stuhldreher et al., 2018) or modeled to construct engine thermal maps (Bera, 2019; Newman et al., 2016; Shourehdeli et al., 2025). Simple but precise enough physics-based tools can then be used to model the entire vehicle's response to a ride for internal combustion engines (Bera, 2019; Kargul et al., 2016, Newman and Dekraker, 2016) or electric engines (Sher et al.,
2021; Kargul et al., 2025). In this study, $\eta_m$ is considered constant and equal to 0.90 as in Bera (2019), since its variations for every driving condition are small compared to the variations of $\eta_e(n, M)$ and an indirect estimate of $\eta_e$ is developed $\overline{\eta_e}(\overline{v})$ that accounts for the driver behaviour as explained in Sect. 2.2 and detailed in Appendix B.

The vehicle dynamics and internal energy balance are modelled through a system of 4 simple equations. The vehicle trajectory is assumed to be a rectilinear motion on a road assumed to be flat. Following the simple tools developed to model the
response of a vehicle, the traction force imposed by the engine on the wheels $F_{trac}$ in newton is calculated from an equilibrium equation applied to the vehicle following the kinematics of point masses. As the other terms of the motion equation can be estimated, this allows to deduce the force, and corresponding power $P_{trac}$. A fraction of the oil consumed by the internal combustion engine of vehicles that deliver the power $P_{fuel}$ in watts is transformed into mechanical power with traction power $P_{trac}$ in watt through the total instantaneous vehicle efficiency coefficient $\eta$. The power lost due to the vehicle mechanical and

thermodynamical inefficiencies is then dissipated as heat $P_{heat\ loss}$ in watts. This system is written:

$$\begin{cases} m\frac{dv}{dt} & = F_{trac} - (F_r + F_{aero}) \\ P_{trac} & = vF_{trac} \\ P_{trac} & = \eta P_{fuel} \\ P_{heat\ loss} & = P_{fuel} - P_{trac} \end{cases} \tag{6}$$

with $v$ (m s$^{-1}$) the vehicle speed, $F_r$ in newton, the rolling friction force, and $F_{aero}$ in newton the aerodynamical drag force. $F_{aero}$ is expressed as usual for vehicle kinematics corresponding to high Reynolds number, and the rolling friction force $F_r$ is expressed following Bera (2019) and Jazar (2009) as an empirical formula valid for a 4-wheel passenger car:

$$F_r = mg(8 \times 10^{-4}(5.1 + \frac{5.5 \times 10^5 + 90mg}{p_t} + \frac{1100 + 0.0388mg}{p_t})v^2) \tag{7}$$

$$F_{aero} = \frac{1}{2}\rho C_d A v^2 \tag{8}$$

With $p_t$ (Pa) the tyre pressure, m (kg) the vehicle mass, g (m s$^{-2}$) the gravity constant, $\rho$ (kg m$^{-3}$) the air density, C$_d$ the drag coefficient and A (m$^2$) the frontal area of a vehicle.

Then three strategies are used to parameterise the heat transferred to the road surface from tyre friction, and the heat released
in the atmosphere. First, in the model, it is assumed that the rolling resistance is fully converted as heat flux to the road surface. Indeed, the constraining forces acting against the vehicle force produced ($F_{trac}$) also contribute to the heat transferred to the environment. The rolling resistance acts as a mechanical constraint on the tyres and the road, part of which is dissipated as heat. Since the tyre temperature, the heat transfer coefficient, the amount of energy transferred to the tyre and the road, and the amount of energy directly converted into heat are unknown, the rolling resistance power $P_r = F_r v$ is fully converted as heat
flux to the road surface.

Then, a part of the power dissipated as heat $P_{heat\ loss}$, calculated with Eq. (6), is used to warm the bottom vehicle body at temperature $T_{veh}$ as explained in Sect. 2.1. It increases the infrared radiation emitted to the road surface $LW_{veh \to rd}^{\downarrow}$ (W m$^{-2}$) because of the warmer bottom vehicle body surface. So, starting from the bottom of the vehicle body in thermal equilibrium with the environment at $T_{can}$ (K), the power $P_{lw}$ (W m$^{-2}$) to reach the infrared radiation emitted at temperature $T_{veh}$ (K) is:

$$P_{lw} = LW_{veh \to rd}^{\downarrow}(T_{veh}) - LW_{veh \to rd}^{\downarrow}(T_{can}) \tag{9}$$

This power is then taken from the total heat produced by the vehicle $P_{heat\ loss}$.

Finally, to take into account the effects of the vehicle system in the TEB model, these effects are then aggregated for the entire traffic with a traffic flow $\Phi$ (vehicles s$^{-1}$) and an average traffic speed $\overline{v}$ (m s$^{-1}$). In addition, the large heterogeneity of vehicle characteristics and behaviour is taken into account in these previous equations with the methodology explained in Sect.
2.2 and described in Appendix B through regression equation estimates $(\overline{v^2}(\overline{v}), \overline{p}(\overline{v}), \overline{\eta_e}(\overline{v}), \overline{a})$ of $\overline{v}$ and vehicle fleet average characteristics $(\overline{m}, \overline{l}, \overline{w}, \overline{A}, \overline{C_d})$.

So, the heat transferred to the road surface from tyre friction is modeled through the rolling resistance power averaged over the vehicle surface $S_{veh}$ (m$^{-2}$) for the entire vehicle fleet $Q_r$ in watt per metre squared written:

$$Q_r = f_{traff} \frac{F_r(\overline{v})\overline{v}}{S_{veh}} \tag{10}$$

In addition, the power dissipated as heat $P_{heat\,loss}$ over each vehicle surfaces $S_{veh}$ (m$^{-2}$) is released in the atmosphere. This power is calculated over the entire vehicle fleet averaged in watt per metre squared $Q_l$ and written in TEB as:

$$Q_l = f_{traff} \frac{P_{heat\,loss}(\overline{v})}{S_{veh}} - f_{traff} P_{lw} \tag{11}$$

$Q_l$ is then transformed as a source of sensible and latent heat flux using the formulation of Pigeon et al. (2007):

$$Q_{Hengine} = 0.92 Q_l \tag{12}$$

$$Q_{Eengine} = 0.08 Q_l \tag{13}$$

These sensible and latent heat fluxes are released in the aggregated heat flux over the entire model tile. This energy is transferred directly to the atmosphere. Thus, the energy released by the traffic can have an effect on the town physical variables only when the model TEB is coupled with an atmospheric model.

### 3.3   Modification of the turbulent heat exchange with the road surface

The collection of vehicles driving on a street immersed in an urban canyon has direct effects on the physical variables of the environment. The physical body of a vehicle is an obstacle that induces drag, increases fluid velocities, and turbulence. Turbulence plays a key role in the boundary layer. It leads to heat and moisture exchange between the air layer and the different surfaces. In this study, only the influence of traffic on the road surface and the lower air layer is considered. Thus, the turbulent exchange coefficient between the road surface and the first air layer is modified by the traffic impact in the model.

Each vehicle is considered to be an independent system. The air motion triggered by a moving vehicle can be described using three regions: along the sides of the vehicle, in its near-wake, and in its far-wake. In the first area, the fluid flow is assumed to be laminar to prevent extensive calculations. From the Navier-Stokes equation, the fluid is assumed to be incompressible, without pressure forces, and to have reached a steady state. Under a vehicle, the fluid is considered to move between two infinite parallel plates with the upper one moving tangentially relative to the other. This flow is modelled as a simple linear Couette flow $U_l$ (m s$^{-1}$) as shown in Fig. 3.

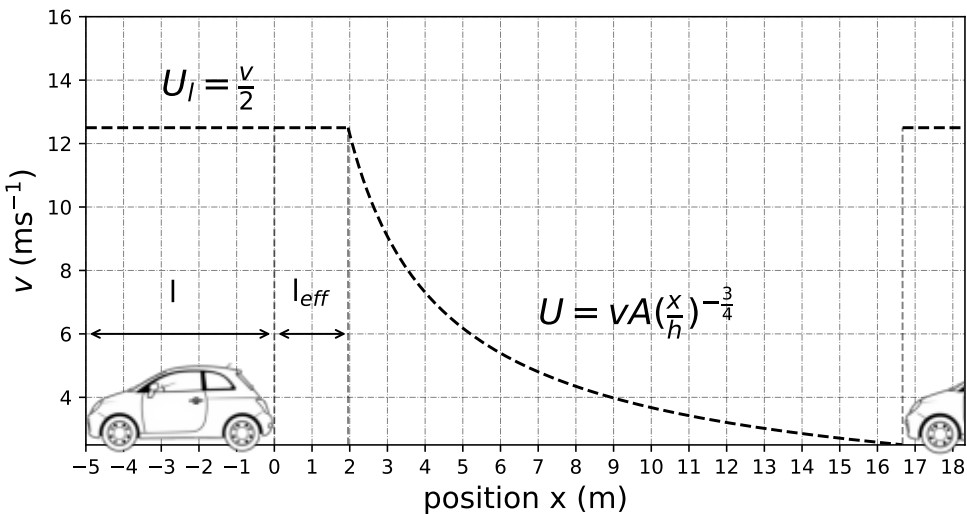

**Figure 3.** Schematic representation of the wind induced by a vehicle and the modelled equations on the urban canyon, with the wind-induced under the vehicle ($l$) on the near-wake ($l_{eff}$) and on the far-wake up to the next vehicle

A simple power law $U \sim x^{-3/4}$ is used to calculate the average wind induced by vehicles in the wake of a vehicle from Eskridge et al. (1979). They developed an analytical formula to calculate the longitudinal velocity field deficit $U$ (m s$^{-1}$) in the wake of a vehicle. They linearised the Navier-Stokes momentum equation using a perturbation analysis. This equation depends on several parameters estimated on wind tunnel experiments. The longitudinal velocity field deficit $U$ is determined
in the absence of crosswinds, stable conditions, and low natural wind velocities and is valid after a downwind distance equal to approximately ten vehicle heights $h$ (Baker, 2001). The reader should refer to Eskridge et al. (1979) for a complete demonstration. The longitudinal velocity in the wake also depends on the other coordinates. It has been used in various studies to calculate the dispersion of pollutants in the wake of a vehicle (Eskridge and Rao, 1983; Hargreaves and Baker, 1997) and improved (Rao, 2002). The vertical and horizontal velocity components can also be calculated according to Hider et al. (1997).
Two assumptions are made to keep the wind-induced formula simple and with consistent mathematical properties: (1) It is assumed that the formula $U(x)$ is also valid in the near-wake of the vehicle (i.e x < 10h). However, a lower bound is determined (i.e x > $l_{eff}$) for continuity reason with the Couette flow $U_l$. Thus, the Couette flow is extended up to $l_{eff}$ in the near-wake of a vehicle, then $U(x)$ is used further in the wake of the vehicle when $U(x) < U_l$. (2) Then, an average wind speed induced by the entire traffic is found $U_{traff}(\overline{v})$ (m s$^{-1}$) by calculating the integral along the x-axis. Each wind induced by a vehicle
is considered to have no overlap from the wind induced by each vehicle. So, the average wind speed is calculated along the vehicle and in the wake until the front of the next vehicle $U_{traff}(\overline{v})$ as shown in Fig. 3. The complete demonstration is given in Appendix A.

This analytical formula $U_{traff}(\overline{v})$ has several refinements over the empirical equation from Fujimoto et al. (2008) and used by Khalifa et al. (2016). The formula $U_{traff}(\overline{v})$ depends explicitly on the vehicle height, length, speed, and traffic intensity, whereas the empirical formula from Fujimoto et al. (2008) depend on the vehicle speed only.

In TEB, the sensible and latent heat between the road surface and the air layer are calculated at ground level, and both use the wind speed at level $z$. So, the wind speed $U_{traff}(\overline{v})$ is vertically interpolated to the level $z$ using the Monin–Obukhov log-wind profile under neutral conditions adjustments as:

$$U_{traff}(\overline{v}, z) = U_{traff}(\overline{v}) \frac{ln(\frac{z}{z_0})}{ln(\frac{z_{traff}}{z_0})} \tag{14}$$

With $z_o$ (m) the road roughness length, $z_{traff}$ the height of the traffic-induced wind. In this study, $z_{traff}$ is set at mid-height of the vehicle. Contrary to Khalifa et al. 2016 , the increased turbulent exchange caused by traffic is not a new component but a direct modification of the turbulent exchange coefficients embedded in TEB. The sum of the wind components at level z gives:

$$U_{eff} = \sqrt{U_{can}^2 + (u^* + w^*)^2 + U_{traff}(\overline{v}, z)^2} \tag{15}$$

With $(u^* + w^*)^2$ (m s$^{-1}$) the total turbulent wind component with $w^*$ (m s$^{-1}$) caused by the local canyon convection and $U_{can}$ (m s$^{-1}$) the natural wind at the lower level. It is used to adjust the aerodynamic resistance of the sensible and latent heat exchange with the road surface $R_{rd}$ calculated as in Lemonsu et al. (2012).

## 4 Experimental set-up and model configurations

This model improvement study is based on the version of the TEB model described in Colas et al. (2024), which models cold conditions using explicit modelling of snow and ice with processes of water melting and freezing. This version of TEB is used as a reference, and compared with the modified version named TEB-CAR, which includes the anthropic processes described in the previous sections. Both models are configured as the TEB-ES version in Colas et al. (2024) except for some changes to the snow removal parameterisation: the total snow cover is removed whenever the snow has been continuously present on the ground for 6 hours, except at night between 0 am and 5 am. Six levels are taken for the surface boundary layer option (Masson and Seity, 2009) for both models with $T_{can}$ (K) the lower air temperature simulated at 0.5 m.

Measurements of long data series on busy road lanes of paired traffic, weather and surface physical variables are essential for this study. Thus, road weather stations from Southern Finland are chosen because they collect in-situ measurements of these variables as shown in Fig. 4. Among the road weather stations deployed on the Turkü-Helsinki highway, Nupuri (60.22805N, 24.59641E) and Palojärvi (60.29328N, 24.31916E) road weather stations are chosen because a strong vehicle commuting pattern is observed at these stations. Indeed, the majority of the commuters go to Helsinki in the morning, then drive back home in the afternoon. This pattern creates clear differences on the road surface physical variables between both directions. It allows to isolate the traffic impact from the other effects since both directions are subject to the same atmospheric conditions. This study takes advantage of this commuting pattern to evaluate the TEB-CAR capacity to model the marginal effects of traffic impacts at these road weather stations.

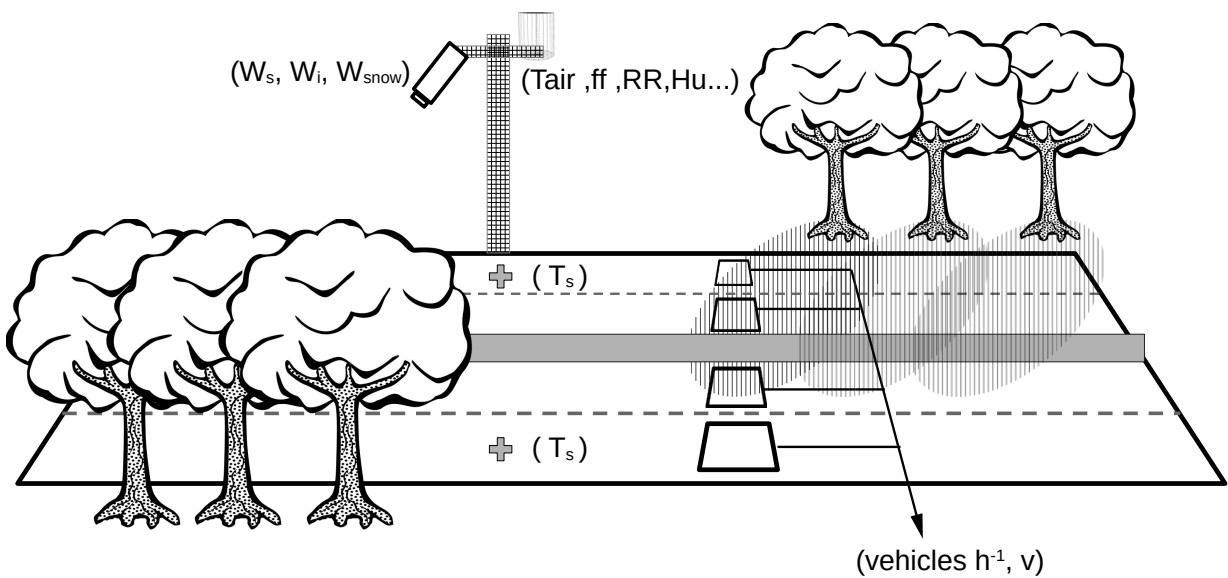

**Figure 4.** Schematic representation of the 2x2 highway between Turku and Helsinki with roadside trees at a weather station location installed by Fintraffic. On the roadside tower, atmospheric sensors are installed with Air temperature (Tair), wind speed (ff), precipitation (RR) and optical surface conditions sensors, ice ($W_i$), snow ($W_{snow}$), water ($W_s$). On the busy lanes, surface sensors measure road surface temperature ($T_s$), and traffic counting systems measure the number of vehicles and the vehicle speed $v$.

These road weather stations manufactured by Vaisala measure common atmospheric variables from roadside towers such as air temperature, wind direction and speed, humidity, precipitation, and also road surface conditions. Water, ice and snow on the road are measured by optical sensors and road surface temperature is measured with asphalt embedded sensors. These physical variables are directly influenced by the effects of traffic and winter maintenance operations with large impacts on the road surface conditions. Road temperature sensors are buried under one high-speed lane for each direction. In addition, Fintraffic
installed a vehicle counting system several kilometres ahead in each lane. These in-situ measurements are transformed and used to force both model versions in this study. They are available in the Zenodo dataset attached to this study (Colas, 2025).

    The road weather stations measure the atmospheric and surface variables every 6 minutes. They are transformed into hourly measurements to force the TEB and TEB-CAR models. To calculate hourly values, the value closest to the full hour is extracted. The 6-minutes accumulated precipitation measurements are aggregated every full hour. Snow and rain are discriminated using
the following criterion: If the air temperature is > 274.15 K, the precipitation is deemed to be liquid, otherwise it is classified as snow as in Colas et al. (2024). The ERA5 reanalysis of the shortwave and longwave data at ground level is also used to force the model by selecting the grid point closest from the Nupuri and Palojärvi locations. Hourly traffic data, composed of vehicle counts and speed, are extracted from the Fintraffic API at the same location. TEB-CAR simulations are run on each direction of the road, with their associated vehicle counts every hour. The traffic counts from the slow lanes, for each direction of the

road, are used to force the models because the probe embedded in the asphalt is located in the slow lanes of the pavement. Thus, in this study, it is assumed that traffic on the faster lanes has no effect on the conditions of the slower lanes.

Simulations are done at both Nupuri and Palojärvi location when atmospheric, surface and traffic observations are available. At Palojärvi location, a simulation of two-month and a half is done between 19 October 2017 and 30 December 2017. A longer simulation is performed at Nupuri location from 19 October 2017 to 1 May 2018. The joint two-month and a half observation period available at both the Palojärvi and Nupuri are used to validate the model at these both locations in Sect. 5.1. Because the surface temperature probe is embedded close to the track lane, the vehicle to road width ratio $\overline{w}/w_{rd}$ is set to 1 in the traffic fraction occupation $f_{traff}$ in this subsection in order to have a road surface temperature modelled that represents the surface covered by the cars and corresponding to the observed one.

In Sect. 5.2, an ablation setup is implemented. It means that for each road direction, 3 more simulations are launched, each with a traffic impact removed from the model. They are called rolling friction, radiative, and wind-induced. The heat released by combustion is not considered for this part because this flux is released on the upper vertical domain of the grid and so have no impact on the road surface temperature. By removing a traffic impact for each simulation, it is possible to investigate the relative impact of each traffic parameterisation on the simulated variables when compared with the reference simulation of TEB-CAR. To evaluate the impact of the traffic on the full road lane width, in this section the vehicle to road width ratio $\overline{w}/w_{rd}$ is set to 0.5 in the traffic fraction occupation $f_{traff}$.

Finally, throughout the simulation period, the traffic counts show that more than 95% of the vehicles driven are passenger cars at both road weather stations (Colas, 2025). Therefore, to avoid complexity, only one vehicle type is considered for the estimation of the traffic parameters, with trucks, buses, and two-wheelers omitted. Estimates of the passenger cars engine efficiency and driver behaviors are made with the corresponding WLTC cycle and manufacturers' data (Colas, 2025). In addition, the missing input traffic parameters for the TEB-CAR simulations (average mass, length, and height of the vehicles) are derived from the ICCT yearly passenger car statistics (ICCT, 2023). In this study, the average vehicle body characteristics of passenger cars sold in 2018 for the EU-28 are taken as input values and are shown in Table 1.

## 5 Evaluation at Nupuri and Palojärvi locations

### 5.1 Performances of the traffic impacts parameterisations

A significant commuting pattern is observed at the Nupuri and Palojärvi road weather stations during the simulation period, with an average of 1200 and 1100 vehicles per hour, respectively, during morning peak hours towards Helsinki and 1000 and 800 vehicles per hour, respectively, in the opposite direction during the afternoon. The time series shown in Fig. 5 gives insights about the traffic-induced impacts throughout the simulation. It is composed of a subset period of 5 working days until the 23 December 2017 characterised by a clear commuting pattern, and of a subset period of 3 nonworking days up to the 26 December 2017 characterised by a similar traffic intensity in both directions.

There is a clear increase in road surface temperature (RST) simulated by TEB-CAR in both directions compared to the TEB simulation. TEB exhibits a significant cold RST bias with respect to the observations, which is corrected by the new processes

| Parameter | Values |
|---|---|
| Vehicle length $\bar{l}$ | 4.3 m |
| Vehicle cross-section $\overline{A}$ | 2.5 m$^2$ |
| Vehicle tire pressure $\overline{p_t}$ | 2.3 bar |
| Vehicle heigth $\overline{h}$ | 1.8 m |
| Vehicle mechanical efficiency $\eta_m$ | 0.90 |
| Vehicle mass $\overline{m}$ | 1500 |
| Vehicle drag coeff. $\overline{C_d}$ | 0.56 |
| Vehicle albedo $\alpha_{veh}$ | 0.75 |
| Vehicle emissivity $\epsilon_{veh}$ | 0.80 |
| Nupuri Turku avg. speed | 27.70 m s$^{-1}$ |
| Nupuri Helsinki avg. speed | 28.03 m s$^{-1}$ |
| Palojärvi Turku avg. speed | 27.70 m s$^{-1}$ |
| Palojärvi Helsinki avg. speed | 27.70 m s$^{-1}$ |

**Table 1.** TEB-CAR traffic parameters and average speeds at the Palojärvi and Nupuri sites. Parameters with overlines are estimated from ICCT (2023)

integrated into the model. Even during the night, when traffic is sparse, the RST is higher due to the strong heating effects throughout the day. This effect can be highly relevant given that dangerous conditions for drivers occur more frequently in
the morning with a lower RST reaching freezing temperatures. In addition, the TEB-CAR simulations accurately follow the observed RST in both directions in Fig. 5 in terms of trend and maximum or minimum.

One should confirm that the bias reduction is not coincidental with other potential biases such as sensor bias. So, the temperature difference caused by the commuting pattern between the two lanes is used to verify the accuracy of the traffic-induced effects modelled in TEB-CAR. In Fig. 5, the TEB-CAR RST differences ($\Delta RST_{dir}$) match the observed $\Delta RST_{dir}$
amplitude. The strongest $\Delta RST_{dir}$ during peak traffic, especially in the morning, are well reproduced by TEB-CAR. The traffic intensity differences are lower in the afternoon than in the morning, leading to a lower observed $\Delta RST_{dir}$. These amplitudes are well reproduced by TEB-CAR.

The RST differences between the two directions are analysed more precisely in Fig. 6. During weekends, traffic intensity differences are smaller as shown in Fig. 5. Lower traffic intensity differences between the two roads directions ($\Delta Traff$) in the
weekends should lead to lower observed $\Delta RST_{dir}$ in the weekends panels of Fig. 6. However, this pattern is less evident for the observed $\Delta RST_{dir}$ than for the simulated $\Delta RST_{dir}$. During working days, the traffic intensity differences are much bigger as the $\Delta RST_{dir}$ for both observed and simulated values. The $\Delta Traff$ distribution is positively skewed at both locations during working days with values mostly between 0 and -250 vehicles h$^{-1}$. Natural factors and road energy inertia have a direct impact

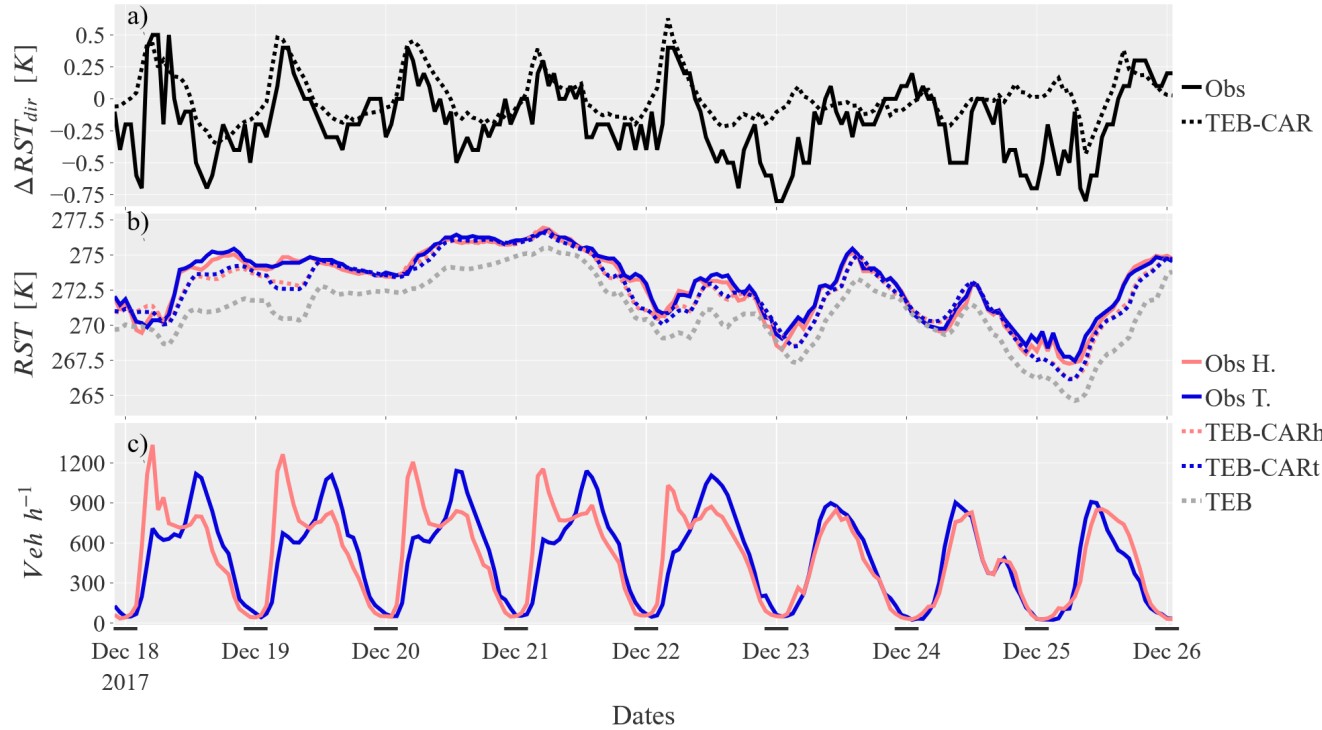

**Figure 5.** Comparison between the observed and simulated RST on a eight days subset period beginning a monday at Nupuri location. The subscripts "h" and "t" are for values on Helsinki and Turku directions respectively. From top to bottom panel : (a) observed and modelled road surface temperature difference between the two road directions ($\Delta RST_{dir}$) (Helsinki minus Turku), (b) road surface temperature, (c) number of vehicles per hour

on the RST with scattered simulated values around the regression line. The RST heteroscedasticity for the higher $\Delta Traff$
suggests that the natural factor and the traffic-induced effects are more intertwined for higher traffic intensities.

Consistency is found between the simulated and observed traffic-induced effects with almost equivalent slopes between the observed and simulated $\Delta RST_{dir}$ from the robust linear regressions (RLM) (Huber, 1973), designed to be robust to outliers. Also, regression equations for both at Nupuri and Palojärvi location have the same slope depending on whether it is calculated on observations or simulations. So, there is a consistent behaviour of the traffic impact on TEB subject to different traffic 390   patterns and atmospheric conditions. The TEB-CAR simulations accurately represent the traffic differences impact up to 750 vehicles per hour. At Nupuri location, the observed and simulated $\Delta RST_{dir}$ distributions are shifted, the intercept of the regression equation for observed $\Delta RST_{dir}$ is negative, and the observed $\Delta RST_{dir}$ have a negative intercept. These features suggest that there is elements that produce this cold bias at Nupuri location that are not taken into account into TEB-CAR. It can also suggests that there is a sensor bias at this location since this effect is not found at Palojärvi location.

Using the RST differences between two road directions subject to the same atmospheric conditions is relevant to extract the traffic impact on the road from natural factors. It also allow to evaluate modelling tools that include traffic impact parameterisation. In addition, the traffic impact parameterisation in TEB-CAR correctly reproduces the traffic impact at the Nupuri and Palojärvi locations with a similar regression equation slope between observed and simulated RST differences.

### 5.2    Analyses of the traffic impacts parameterisations

The cumulative effect of the new set of traffic parameterisations in TEB (named TEB-CAR) results in marked impacts on the temperature of the road. Each traffic parameterisation may have opposite or cumulative effects on the physical variables. In addition, each impact may change depending on atmospheric conditions, seasonality, and traffic intensity. Thus, in this section, the individual effect of each traffic impact in the model is studied using the Nupuri experiment throughout the entire simulation period. A total of eight more simulations at Nupuri are analysed here. Each simulation has a traffic impact removed from the 405    model and is launched in both road directions.

To evaluate the relative contribution of each process on the physical variables, one must compare each simulation launched with a process removed, with the TEB-CAR simulation. Each process has a different marginal impact on the RST and the lower air temperature simulated at 0.5 m $T_{can}$ (K) as shown by Fig. 7 and Fig. 8. The figures show that the RST simulated without the wind-induced parameterisation is warmer than the one with the entire set of traffic impact (TEB-CAR simulation). 410    Thus, it has a cooling effect on the RST because of the stronger heat exchange between the air temperature and the RST. The same trend is observed on the simulated RST with the radiative process removed. However, the cooling effect of the ablated wind-induced simulation is much stronger than for the ablated radiative simulation, in both seasons. This strong impact from the wind-induced parameterisation can be explained by the analysis of the processes Fig. 9: the wind induced by the vehicles accounts for 75% on average of the total wind simulated by the model in the daytime. At night, this proportion falls between 415    15% and 45% on average due to much lower traffic and leads to a lower cooling effect. In spring, the cooling effect of the wind-induced impact is greater than in winter with up to around 2.5 K on average during peak traffic (13 h UTC), as shown in Fig. 8. In addition, the Fig. 8 shows that the standard deviation of the road surface temperature differences with the reference simulation $\Delta RST_{mod}$ is the lowest with the wind-induced parameterisation removed. In fact, the cooling effect of the wind-induced process is more pronounced when the temperature difference between the road surface and the air is the highest. This 420    effect is highly dependent on meteorological conditions and then explains most of the traffic impact variability.

The radiative and the wind-induced processes have an opposite effect on the air temperature simulated at the lowest level $T_{can}$. As shown in Fig. 7, wind-induced leads to a slight warming effect whereas radiative impact leads to a slight cooling effect in both seasons. On average, these effects are greater in the daytime. Since the RST is almost always warmer than the air temperature, more energy is transferred when the differences in soil-air temperature are higher, especially in the daytime. 425    In addition, the radiative impact leads to a slight cooling effect on $T_{can}$ like for the RST.

Furthermore, the RST simulated with the rolling friction is cooler than the RST simulated by TEB-CAR. Thus, this parameterisation has a warming effect on the RST. The warming effect of the heat lost by vehicles is low, even during peak traffic (4 h UTC and 13 h UTC) throughout the period, as shown in Fig. 8. Indeed, the analysis of the processes in Fig. 9 shows the power

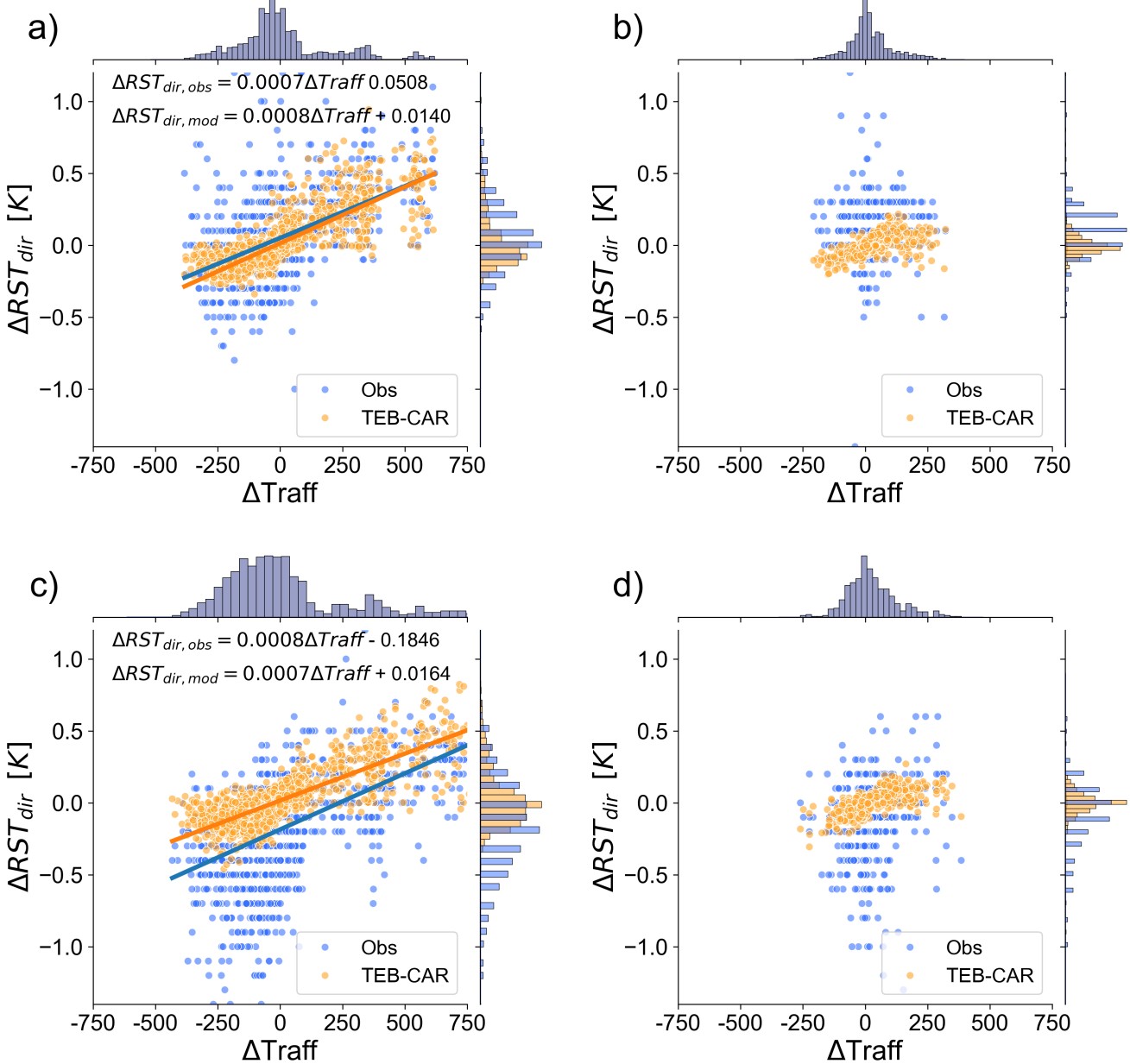

**Figure 6.** Road surface temperature differences $\Delta RST_{dir}$ of both measured and simulated values between Helsinki and Turku directions (Helsinki minus Turku). Robust linear regression lines (RLM) are drawn between $\Delta Traff$ and $\Delta RST_{dir}$ for both observations and simulations. Panels (a) and (b) show the differences at the Palojärvi site on weekdays (a) and weekends (b). Panels (c) and (d) show the same information at the Nupuri site.

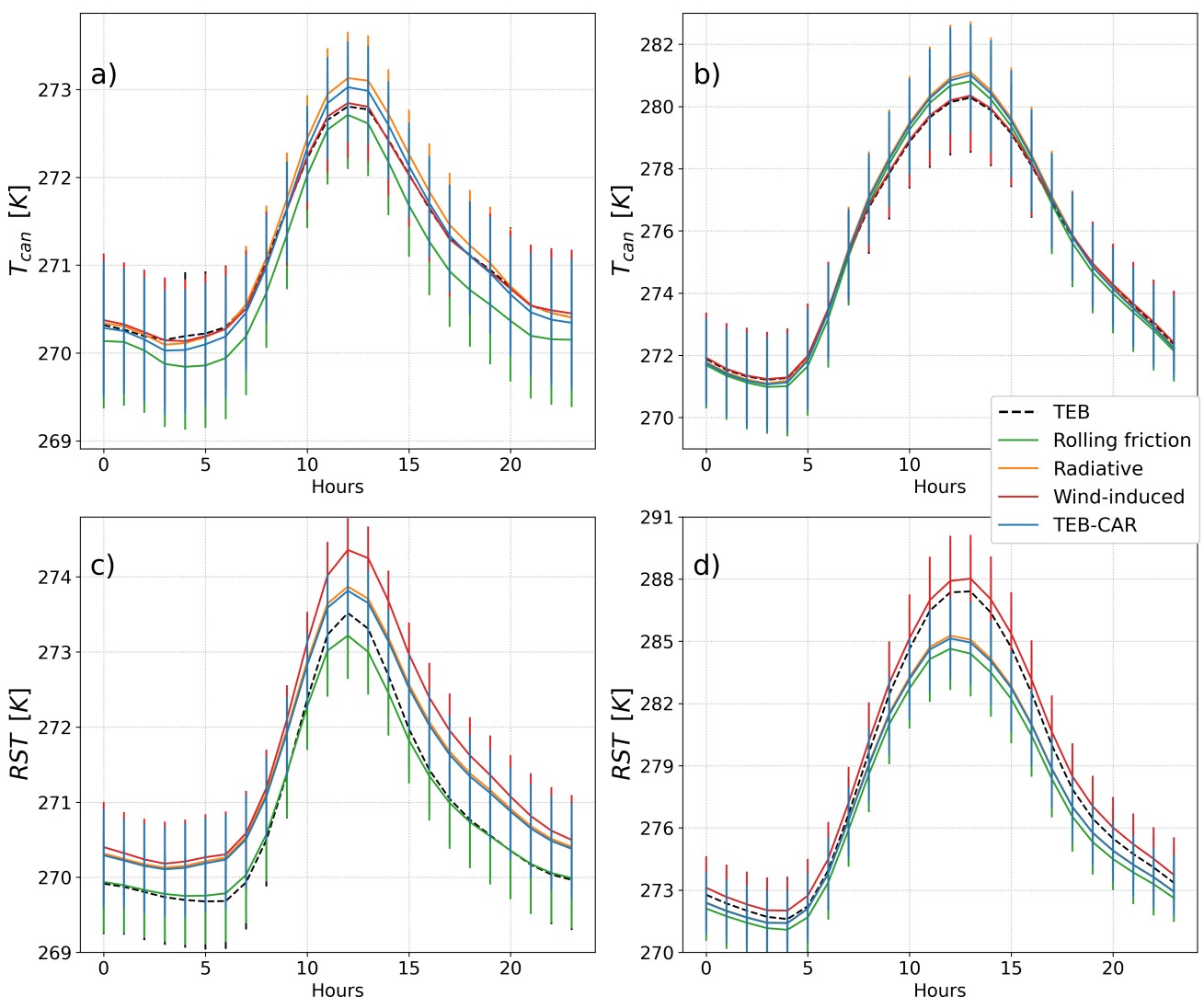

**Figure 7.** On the Turku road direction, TEB, TEB-CAR and ablation experiment simulations with one traffic impact removed for each simulation. Simulated low air temperature $T_{can}$ (K) on panels (a) and (b) and road surface temperature RST (K) at panel (a) and (b) with the confidence interval of the estimator of the expected value. (a) and (c) are calculated on the winter period and (b) and (d) on the spring period

produced by rolling friction and transferred to the road surface is approximately 30 W m$^{-2}$ on average during peak traffic for both road directions. This leads to a strong impact on the RST as shown by the significant shifts in descriptive statistics from the TEB-CAR simulation on the boxplot diagrams in Fig. 8. The source of energy from the rolling friction leads to an increased air temperature simulated at the lowest level $T_{can}$ as shown in Fig. 7. This behaviour is consistent throughout the day and throughout both seasons.

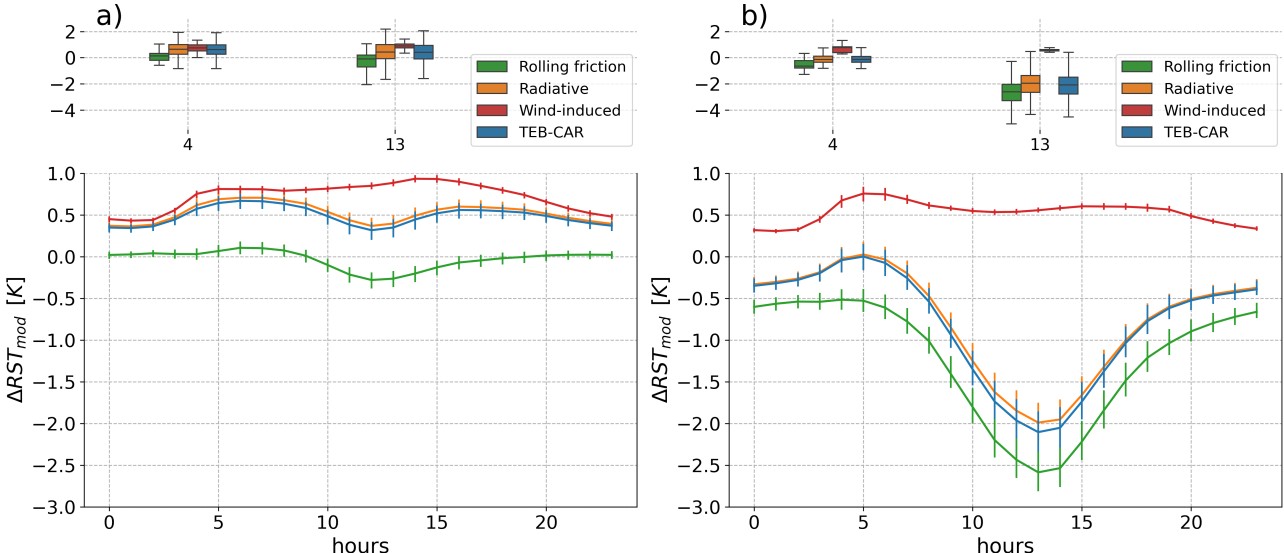

**Figure 8.** On the Helsinki road direction, road surface temperature differences $\Delta RST_{mod}$ between TEB-CAR, the ablation experiment simulations with TEB (TEB-CAR minus TEB). Boxplots are drawn in peak traffic (4:00 h UTC and 13:00 h UTC) on the upper panels and lines of the average hourly values on the bottom panels. $\Delta RST_{mod}$ are calculated for the winter period in (a) and spring in (b) with the confidence interval of the estimator of the expected value.

To summarize, the processes added in TEB-CAR each have a different marginal impact on the RST: rolling friction has
435 a strong marginal heating effect, increased turbulence a strong marginal cooling effect and radiative effect a small marginal cooling effect. In addition, depending on the meteorological conditions, the impact of traffic can change significantly due to the temperature dependency of the wind-induced parameterisation. The cumulative effect of the new set of traffic parameterisations in TEB (named TEB-CAR) results in marked differences compared to the model without the traffic parameterisation. In both seasons, the overall impact on the RST is driven by the competition between the wind-induced impact and rolling friction.
Competition between these factors eventually leads to an overall warming effect on the RST in winter and a cooling effect in spring in both direction as shown on Fig. 7 and Fig. 8. The simulated TEB-CAR RST is 0.5 K warmer during the winter period and is 0.9 K cooler in spring than the RST of TEB. For air temperature $T_{can}$, in both directions, TEB-CAR compared to TEB simulations lead to equal air temperature in winter and and 0.14 K higher on average in spring.

Even if there is no direct impact of the heat released by the fuel combustion on the road surface temperature, it is interesting
to look at the values. The total heat loss by the vehicle inefficiencies modelled in this study is comparable to the heat loss modelled in other studies. Pigeon et al. (2007) calculate 18.3 W m$^{-2}$ released in the atmosphere for 1400 vehicles h$^{-1}$ from the inventory approach in Toulouse city. In Fig. 10, the heat released by traffic is calculated considering the same traffic intensity (1400 vehicles h$^{-1}$) spread over the same area 100m x 100m. Depending on the average speed of road traffic in cities, this study simulates an average heat released from 16 W m$^{-2}$ for 4 m s$^{-1}$ to 6.3 W m$^{-2}$ for 15 m s$^{-1}$.

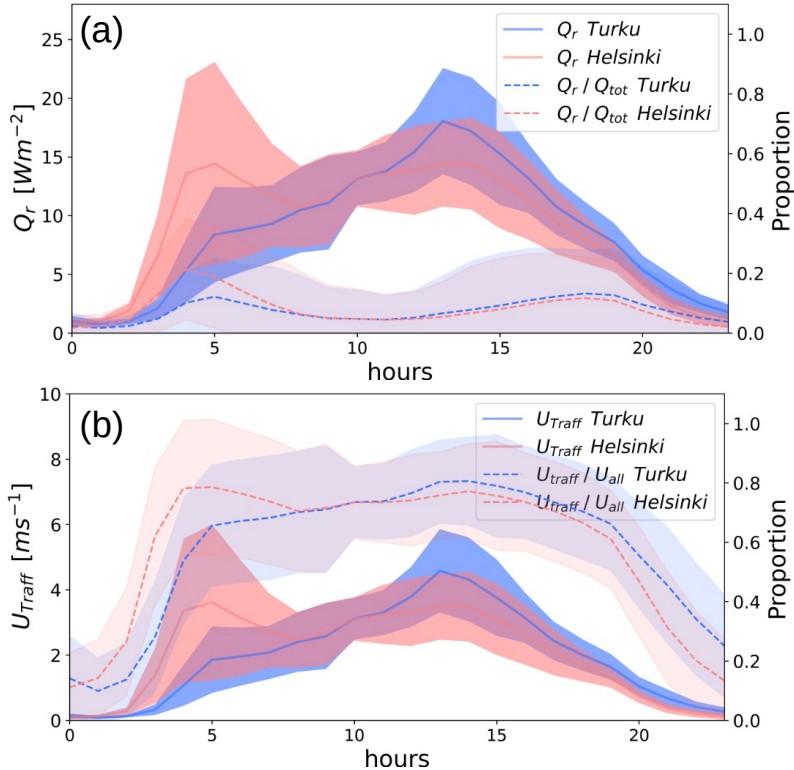

**Figure 9.** Diurnal traffic-induced effects in the TEB-CAR model on the whole simulation period for the Nupuri experiment and for both road directions, with the proportion of the total impact. (a) Rolling friction and its marginal amplitude on the total net heat at the road surface $Q_{tot}$ (b) Wind-induced and its marginal amplitude on the total wind on the lower layer of the atmosphere $U_{all}$ at 0.5 m.

## 6    Discussion and conclusion

In this study, we introduced and evaluated a new modelling strategy to account for traffic-induced impacts in the SURFEX-TEB V9.0 urban climate model. The approach integrates parametrizations for heat released from engine inefficiencies, vehicle body impacts on the radiation budget, turbulent heat exchange, and surface-tyre interactions, all modelled as coherent analytical solutions dependent on vehicle counting. The heterogeneity of driving behaviours and vehicle models was also considered, enhancing the model's realism. The modified model, termed TEB-CAR, was evaluated against observations from two road weather stations in southern Finland, Nupuri and Palojärvi, which exhibit strong commuting patterns. This setup has allowed to extract the traffic-induced effects from other environmental factors, as both road directions experience similar atmospheric conditions. Finally, we analysed the marginal impact of each traffic impact parameterised in the model.

This study demonstrates that traffic has a significant impact on road surface temperature (RST), even for a road with a medium traffic intensity as found in Khalifa et al. (2016). If one can measure the road conditions of two or more roads lanes with different traffic patterns, it is possible to extract the traffic-related impacts from natural factors. This methodology allows

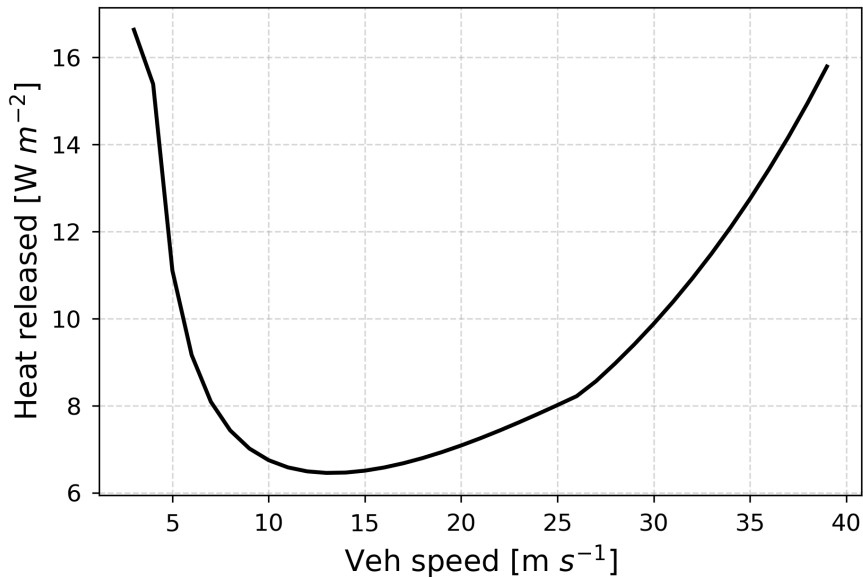

**Figure 10.** Power lost in the air from fuel combustion in a grid of 100 x 100m by the vehicles for a traffic intensity of 1400 veh h$^{-1}$ for the Finland location

to evaluate the parameterisations of the traffic impacts in models as in road weather forecast or urban climate. TEB-CAR significantly improves the simulation of the road surface temperature (RST) compared to the reference TEB model. Compared to the observed RST differences between both directions at Nupuri and Palojärvi location, TEB-CAR is able to reproduce the observed trend caused by differences in traffic intensity. In addition, depending on atmospheric conditions, magnitude, timing, and trend of road traffic, TEB-CAR modify the physical variables. TEB-CAR simulates an increased RST for cold air temperatures and low downward solar fluxes and a decreased RST for warm air temperatures and high downward solar fluxes. A RST of several degrees higher in cold conditions can significantly influence the forecast of dangerous road conditions.

This study shows that taking into account the full set of traffic impacts is relevant to simulate road conditions and atmospheric physical variables, as corroborated by Khalifa et al. (2016). In particular, the competition between the two major effects, the rolling resistance and the wind-induced by the vehicles is the most important characteristic of the traffic parameterisation in this study. In contrast, the impact of the heat lost from fuel combustion is not directly studied because a atmospheric model is needed to simulate the energy feedback between the atmosphere and the land variables. As shown in this study, simulating the impact of road traffic on local climate with a simple aggregated source of heat released in the atmosphere (Bohnenstengel et al., 2013; Pigeon et al., 2007; Iamarino et al., 2011) may not be enough to capture the full extent of traffic impacts. This estimate calculated by Pigeon et al. (2007) or by Sailor and Lu (2004) is of the same order of magnitude as the one calculated in this study and described in appendix B.2.

Some road traffic impacts are overlooked in this study. The current traffic-induced effects do not take into account road conditions, such as water, ice, slush, or snow, which could significantly alter the dynamics of surface-tyre interactions. Also, the study did not explicitly consider the impact of heavy vehicles, which despite being a small percentage of the traffic could have a significant impact on RST due to their larger size and weight. It could be defined as a second vehicle type with its own set of estimated characteristics. The traffic impacts of the set of two average vehicle types could then be calculated as a simple arithmetic mean, weighted by a vehicle type ratio. Finally, the traffic intensity of the Helsinki-Turku highway is moderate compared to that observed on the main urban ring roads (Amato et al., 2016). Future work should evaluate the model subject to higher traffic intensities and summer conditions to assess the reliability of the traffic impacts parameterisation. The simulation period studied was relatively short, on winter and spring conditions only and only two road weather stations have been used to assess TEB-CAR improvements. A more thorough evaluation would provide a stronger confidence in these parameterizations and in the model performance across different traffic patterns. It would also be relevant to assess the overall impact of traffic on the urban climate.

Despite these limitations, TEB-CAR represents a significant step forward in explicitly taking into account the traffic-induced impacts on road surface conditions and on physical variables of the atmosphere. The ability of the model to capture the impact of traffic, particularly during peak commuting hours, has the potential to improve road safety and maintenance operations in winter. Moreover, the TEB-CAR version has shown strong improvements in simulating road surface temperatures. Future research should focus on refining the parameterisations, extend the evaluation to more diverse and high-traffic environments, and to include additional factors such as the direct traffic effects on slippery conditions. This would further validate the robustness of the model and increase its applicability to different settings.

**Appendix A: Detailed calculations of the wind induced by the traffic**

An analytical formula is found to model the wind induced by the traffic $U_{traff}$ which is integrated into the TEB model. To model the fluid velocities induced by a vehicle, three areas are considered: along the length of a vehicle, in the near-wake of a vehicle, and in the far-wake of a vehicle.

First, the velocity of the fluid under the vehicle is determined. The fluid is assumed to be incompressible, the pressure forces are considered negligible, and the steady state is found so that the partial derivative of the fluid speed $U_l$ to time is null. Under a vehicle, the fluid is considered to be between two infinite parallel plates with the upper one moving tangentially relative to the other. The Navier-Stokes momentum equation simplify to:

$$\frac{d^2 U_l}{dz^2} = 0 \tag{A1}$$

For $z = 0$ the road surface and the boundary conditions $U_l(0) = 0$ and $U_l(h) = v$ with $v$ the vehicle speed. The exact solution gives:

$$U_l = v \frac{z}{h} \tag{A2}$$

So, the average wind speed between the road surface and the bottom of the car body at the height $h$ yields:

$$U_l = \frac{v}{2} \tag{A3}$$

Then, the formula from Eskridge et al. (1979) is used to determine the wind speed produced in the wake of a vehicle. Eskridge et al. (1979) determined that beyond the recirculation region (x < 10h) and for (u, v, w $\leq$ v$_{car}$) it is possible to linearise the Navier-Stokes momentum equation with the perturbation analysis. It could be possible to model with an explicit formula the near-wake described by large-scale flow structures with high instabilities of the vehicle as the jet-plan turbulent flow power law $U \sim x^{-1/2}$ as defined in Youssef (2012). The exact solution of the longitudinal wind speed deficit in the wake of a vehicle gives for the maximum value of the wind $U$ at a given value of x is:

$$U = vA\left(\frac{x}{h}\right)^{-\frac{3}{4}} \tag{A4}$$

$$A^{'} = \left(\frac{C^{'}}{\gamma^3 h^2 v^2 (32\pi)^{\frac{1}{2}} \Lambda}\right)^{\frac{1}{4}} \tag{A5}$$

$$C^{'} = -C = \frac{1}{2}\rho v_{vh}^2 C_D Ah \tag{A6}$$

with $A^{'}$ a constant, $C_D$ the drag coefficient, $h$ (m) the height of the vehicle, A (m$^2$) the cross-sectional area in the direction of motion, $C^{'}$ the flow couple on the vehicle, $\gamma = 0.4$ and $\Lambda = 4.13$ two coefficients estimated in Eskridge et al. 1979, $\rho$ (kg m$^{-3}$) the density of the air. Refinements are possible by taking into account the other coordinates from the Eskridge et al. (1979) formula, the refined formula from Eskridge et al. (1982) or from Hider et al. (1997). In addition, the following assumption is considered: the wind-induced by the vehicle has an effect only on the width of the vehicle width. Thus, when estimating the wind induced by the vehicle on the entire lane width, Eq. (A4) is modified as:

$$U = \frac{w}{w_{rd}} vA\left(\frac{x}{h}\right)^{-\frac{3}{4}} \tag{A7}$$

with the factor $w/w_{rd}$ with $w$ the mean vehicle width and $w_{rd}$ the road width, which allows to get the average impact on the road lane dimensions.

Then, the average wind speed induced by the total traffic is modelled considering no overlap from the wind induced by each vehicle. The average wind speed is calculated along the vehicle and in the wake until the front of the next vehicle $U_{traff}$. To keep the formula consistent, it is assumed that the formula $U(x)$ is also valid in the near-wake of the vehicle (i.e x < 10h). However, a lower bound is determined (i.e x > $l_{eff}$) for continuity reason with the Couette flow $U_l$. Thus, the Couette flow is extended up to $l_{eff}$ in the near-wake of a vehicle, then $U(x)$ is used further in the wake of the vehicle when $U(x) < U_l$. Thus, the length $l_{eff}$ (m) calculated to keep the values of the wind speed within the limit $U(l_{eff}) < \frac{v}{2}$ is written as:

$$l_{eff} = \frac{h}{(2A)^{-\frac{4}{3}}} \tag{A8}$$

Then, an average wind speed induced by the entire traffic $U_{traff}$ (m s$^{-1}$) is found by calculating the integral along the x-axis. The average wind speed induced by the vehicle fleet with $U_{max}$ the wind induced behind the vehicle $U_z$ the wind induced under the vehicle is expressed as:

$$U_{traff} = \frac{w}{w_{rd}} \frac{1}{(\frac{v}{\phi} + l + l_{eff})} \left( \int_{-l}^{l_{eff}} U_l \, dx + \int_{l_{eff}}^{v/\phi} U(x) \, dx \right) \tag{A9}$$

$$U_{traff} = \frac{w}{w_{rd}} \frac{1}{(\frac{v}{\phi} + l + l_{eff})} \left( \frac{v}{2}(l + l_{eff}) + 4Ah^{\frac{3}{4}}v((\frac{v}{\phi})^{\frac{1}{4}} - l_{eff}^{\frac{1}{4}}) \right) \tag{A10}$$

This formula has satisfactory boundary condition with $\lim_{v \to 0} U_{traff}(v) = 0$.

Finally with the heterogeneous driving conditions, we compute the average wind speed according to the underlying distribution of the vehicle speed. So for the average vehicle fleet speed $\overline{v}$ and the Monte-Carlo estimate $\overline{v^{\frac{1}{4}}}$ explained next section, we get:

$$U_{traff}(\overline{v}) \simeq \frac{\overline{w}}{w_{rd}} \frac{1}{(\frac{\overline{v}}{\phi} + l + l_{eff})} \left( \frac{\overline{v}}{2}(l + l_{eff}) + 4Ah^{\frac{3}{4}}\overline{v}(\overline{v^{\frac{1}{4}}}(\phi)^{-\frac{1}{4}} - l_{eff}^{\frac{1}{4}}) \right) \tag{A11}$$

In TEB, the sensible and latent heat fluxes between the road surface and the air layer are calculated at height $z$ depending on the wind speed at the same level. Thus, the wind speed $U_{traff}(\overline{v})$ is modified by a vertical interpolation to height $z$ by assuming a Monin–Obukhov log-wind profile under neutral conditions. First, the equality between Eq. (A11) and the log-wind profile at height $z_{traff} = \overline{h}/2$ is given as:

$$U_{traff}(\overline{v}) = \frac{u^*}{\kappa} ln(\frac{z_{traff}}{z_0}) \tag{A12}$$

The $u^*/\kappa$ ratio is found with this previous formula and allow to calculate the wind induced by the traffic at height $z$ with the same log-wind profile. Finally at height $z$ the wind induced by the vehicle gives:

$$U_{traff}(z) = U_{traff} \frac{ln(\frac{z}{z_0})}{ln(\frac{z_{traff}}{z_0})} \tag{A13}$$

## Appendix B: Estimation of the engine efficiency and fuel consumption of vehicles based on the WLTC dataset

### B1   Method

To estimate the amount of energy lost from fuel combustion in a vehicle engine, one must consider the driver behaviour and the engine response. The WLTC dataset and an automobile fleet characteristics database are exploited. Four passenger car subcycles are considered, indexed by $s = \{1,2,3,4\}$. They are characterised by low-speed, medium-speed, high-speed, and extra-high-speed regimes representative of different road speeds. First, the engine response is estimated in real-world scenario thanks to a databank of vehicle fuel consumption and characteristics and to the WLTC standard on the 4 subcycles. Four average vehicle engine efficiencies are estimated corresponding to the 4 subcycles. Second, thanks to the WLTC standard, estimates of average driver behaviours are calculated with monte-carlo estimators corresponding to the 4 subcycles. Finally, both vehicle engine efficiencies and estimates of average driver behaviours are extended for every possible vehicle fleet average speed $\overline{v}$ (m s$^{-1}$). A simple interpolation is carried out to get values between the 4 subcycles with multiple linear regression.

In France, the consumption of each commercially available car model is documented in a database managed by the Agence De l'Environnement et de la Maitrise de l'Energie (ADEME). This study uses a homogeneous database of vehicles sold between 2023 and 2024 as provided by Colas et al. (2025). Each vehicle has been driven through the WLTC cycle. The vehicle engine provides the force needed to counter the drag forces along the trajectory. The simple Newtonian law of motions models the different forces at stake with the standard equilibrium equation applied to the vehicle. Vehicles in the ADEME database include the latest vehicle models equipped with fuel saving technologies such as start and stop, and fuel injector cut-off.

The efficiency of the vehicle engine is strongly dependent on the engine engineering and differs from one engine to another. We could not access enough manufacturer data to accurately estimate the efficiency of all types of engines, so an indirect estimate of $\eta_e$ was developed. The mechanical efficiency of the vehicle $\eta_m$ is considered constant, since its variations for every driving condition are small compared to the variations of $\eta_e$.

The mean engine efficiency is computed. The start and stop technology turns off the vehicle engine when it is idle. The fuel injectors cut-off suppress the fuel consumption when the accelerator pedal is released. The traction force $F_{trac}(v_i)$ provided by a vehicle engine at each time step $i$ can be broken down as:

$$
\begin{cases}
F_{trac}(v_i) & = F_r(v_i) + F_{aero}(v_i) + ma_i \qquad v_i \geq 0 \\
F_{trac}(v_i) & = 0 \qquad v_i < 0
\end{cases}
\tag{B1}
$$

where $F_r$ and $F_{aero}$ are defined as in Eq. (7) and $a_i$ (ms$^{-2}$) is the instantaneous acceleration. The parameters for the drag coefficient and the cross section area of the vehicle that are missing are inferred from Kukwein (Kühlwein, 2016). Other methods could be used to estimate these parameters when missing, such as the one in Komnos et al. (2024). For each vehicle along a WLTC cycle $s$, there exists a vehicle engine efficiency $\eta_{se}$ that is a key variable that represents the energy lost by the vehicles as defined by the system of equations Eq. (6). The vehicle engine efficiency can be computed as:

$$
\eta_{se} = \frac{\frac{1}{n}\overline{v}_s \sum_{i=1}^{n} F_{trac}(v_i)}{\eta_m P_{fuel}}
\tag{B2}
$$

$$
= \frac{1}{n\eta_m F_{fuel}}\left(m \sum_{i=1}^{n} \frac{dv_i}{dt} + \sum_{i=1}^{n} F_r(v_i) + \sum_{i=1}^{n} F_{aero}(v_i)\right)
\tag{B3}
$$

$$
\tag{B4}
$$

With $m$ (kg) the car mass of a vehicle, $\eta_m$ the mechanical efficiency of a vehicle set to 0.90 consistent with the estimates in Bera (Bera, 2019), $n$ the number of measures along the subcycle $s$, $\overline{v}_s$ the average vehicle speed along the subcycle $s$, and $P_{fuel}$ the total power generated by the fuel consumption of a vehicle written as:

$$
P_{fuel} = \overline{v}_s \lambda_{fuel} \rho_{fuel} C
\tag{B5}
$$

With $\lambda_{fuel}$ the heat of combustion in joule per kilogram, (43.8 10$^6$ J kg$^{-1}$ for gasoline as in Pigeon et al. (Pigeon et al., 2007) and 41.0 10$^6$ J kg$^{-1}$ for essence), $\rho_{fuel}$ (kg m$^{-3}$) the fuel density given as 850 kg m$^{-3}$ in this study and C (m$^3$m$^{-1}$) the fuel vehicle consumption.

This previous calculation is performed for each vehicle and then averaged to give an averaged engine efficiency of the automobile fleet given a specific subcycle $\overline{\eta_{se}}$. This value has two meanings. First, it represents the average vehicle engine efficiency of the total automobile fleet in the vehicle databank, but it also represents the average vehicle engine efficiency of the total automobile fleet at a given average speed $\overline{v}$ under real conditions. Indeed, we can assume that there is an underlying distribution of accelerations and speeds at each location. Each WLTC cycle has been built from a speed and acceleration data sample of the world's driving habits (Tutuianu et al., 2015). So there are two random variables X and Y in $\mathbb{R}$ for each WLTP subcycle of unknown probability density of speed $f_{X,s}$ and acceleration $f_{Y,s}$ such as $X \sim f_{X,s}(v)$ and $Y \sim f_{Y,s}(v)$ and $v$ a parameter that is the average speed of the automobile fleet. Both random variables are considered independent within a WLTC subcycle. Speed and acceleration are also considered independent of the vehicle characteristics vector. Each WLTC subcycle is a Monte Carlo sampling of the underlying probability density of speed $f_{X,s}(U_k)$ and acceleration $f_{Y,s}(U_k)$ with $U_k$ a suitable unknown Monte Carlo estimate. Then, the Monte Carlo samples are used to estimate the variables needed to estimate the traffic impact and to estimate the engine efficiencies $\overline{\eta_{se}}$ that depend on the driver behaviours.

Thus, $\overline{\eta_{se}}$ computed previously is an estimate of the average engine efficiency of the automobile fleet at a given average speed $\overline{v_s}$. The average speed $\overline{v_s}$ of the automobile fleet is computed from the Monte Carlo sampling as:

$$\overline{v_s} = \frac{1}{n} \sum_{k=1}^{n} X_{ks} \tag{B6}$$

Other estimates are needed to compute the average characteristics of the automobile fleet given the estimate of the expected value of $X$ for a given subcycle $s$:

$$\overline{v_s^2} = \frac{1}{n} \sum_{k=1}^{n} X_{ks}^2 \tag{B7}$$

$$\overline{v_s^{\frac{1}{4}}} = \frac{1}{n} \sum_{k=1}^{n} X_{ks}^{\frac{1}{4}} \tag{B8}$$

$$\overline{p_s} = \frac{1}{n} \sum_{k=1}^{n} \mathbb{1}(Y_{ks} \geq 0) \tag{B9}$$

$$\overline{a_s} = \frac{1}{\overline{p_s}n} \sum_{k=1}^{\overline{p_s}n} Y_{ks} \mathbb{1}(Y_{ks} \geq 0) \tag{B10}$$

With, $\overline{v_s^2}$ the average estimate for the squared speed, $\overline{v^{\frac{1}{4}}}$ the average estimate for the power $\frac{1}{4}$ of the speed, and $\overline{p_s}$ the fraction of the total automobile fleet with a positive or null acceleration, $\overline{a_s}$ the average estimate of the positive acceleration. Indeed, the negative acceleration does not contribute to the total force generated by the engine from Eq. (B1).

These terms are then extended for any given average speed $\overline{v}$. Regression equations are learnt to estimate the behaviour of the automobile fleet given the average speed $\overline{v}$. Simple multiple linear regressions (MLRs) are performed and give the estimates $\overline{v^{1/4}}(\overline{v})$, $\overline{v^2}(\overline{v})$, $\overline{p}(\overline{v})$ and $\overline{\eta_e}(\overline{v})$ drawn in Fig. B1. The positive acceleration $\overline{a_s}$ is averaged as a single value $\overline{a} = 0.28$ since no simple relationship with the average speed $\overline{v}$ can be found. To keep the consistency with the dynamic of a real vehicle, two

thresholds are added to these estimates at really low speed and high speed. The following conditions are satisfied:

$$\overline{p}(\overline{v}) \leq 0.54 \tag{B11}$$

$$\overline{\eta}(\overline{v}) \geq 0.08 \tag{B12}$$

The estimates of these variables are consistent with the physics of a vehicle motion. It is then included in TEB to compute the heat from the rolling resistance Eq. (7) and heat released in the air Eq. (11). The modelled key engine efficiency variable is comparable to the engine efficiency observed in a single vehicle (Kargul et al., 2016).

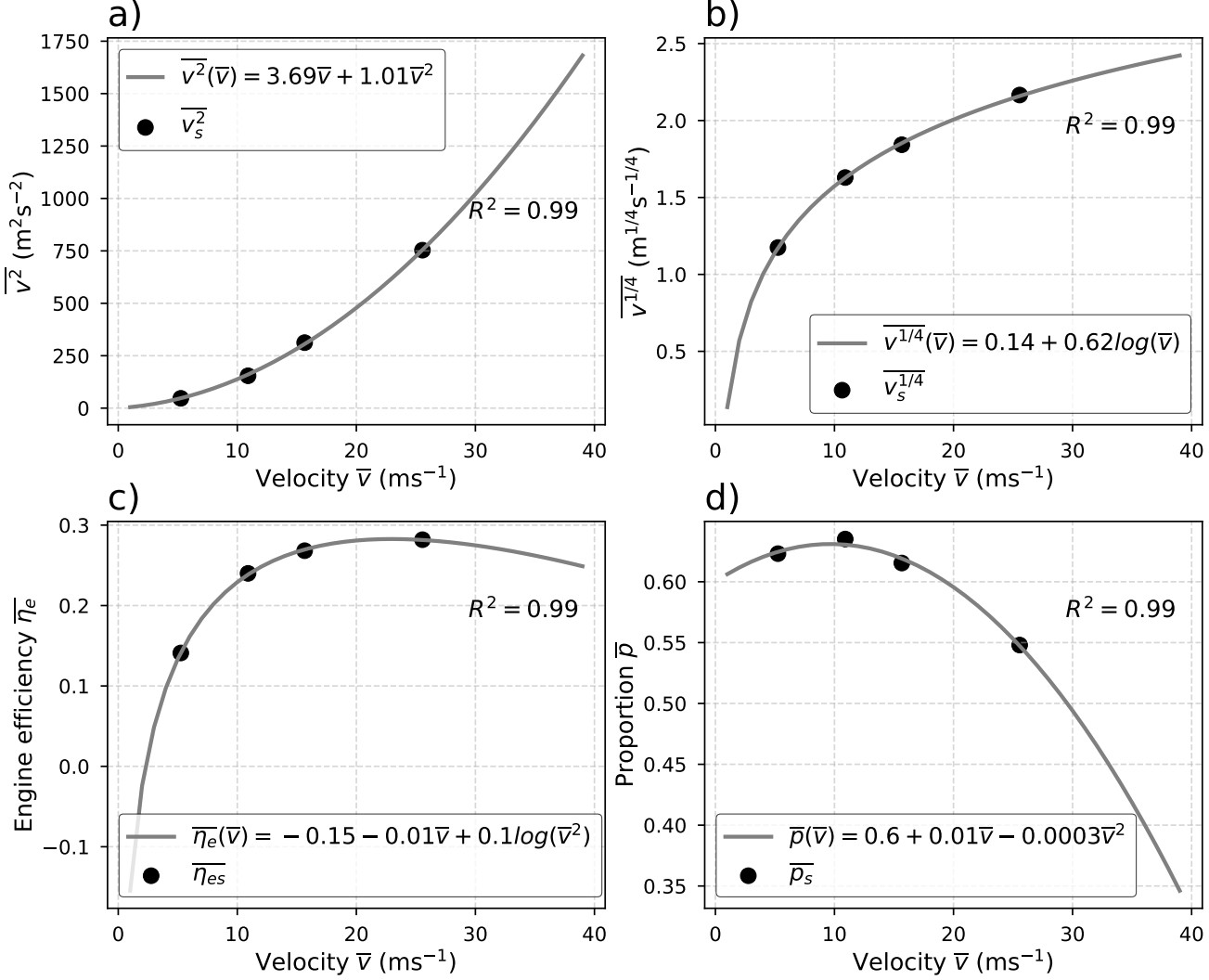

**Figure B1.** Regression equations from the multiple linear regressions (MLRs) of the parameters against the local estimate for each WLTC cycle with the average: (a) squared speed $v_s^2$, (b) speed to the power $1/4$, (c) engine efficiency, (d) acceleration proportion

## B2   Evaluation

The method of estimating engine efficiency and other parameters related to fuel consumption is evaluated against the vehicle dataset. Since there is no direct measurement of the estimated variables, the fuel consumption of each vehicle from the WLTC subcycle is compared to the fuel consumption estimated as:

$$P_{fuel}(\overline{v}) = \frac{1}{\overline{\eta}(\overline{v})\eta_m} (F_r(\overline{v^2}(\overline{v})) + F_{aero}(\overline{v^2}(\overline{v})) + m\overline{a})\overline{p}(\overline{v})\overline{v} \tag{B13}$$

Inside this previous equation, every parameter needed to compute the heat released by a vehicle is used. This estimate can be used to retrieve the fuel consumption by averaging over a diversity of vehicles. This method allows to estimate the values of the parameters that have a significant impact on the heat lost by the vehicles.

The previous method is compared against a simple baseline without estimating the posterior parameters. This baseline is the average fuel consumption of the entire vehicle databank for each WTLC subcycle. Then it is tested against each vehicle fuel consumption in Fig. B2.

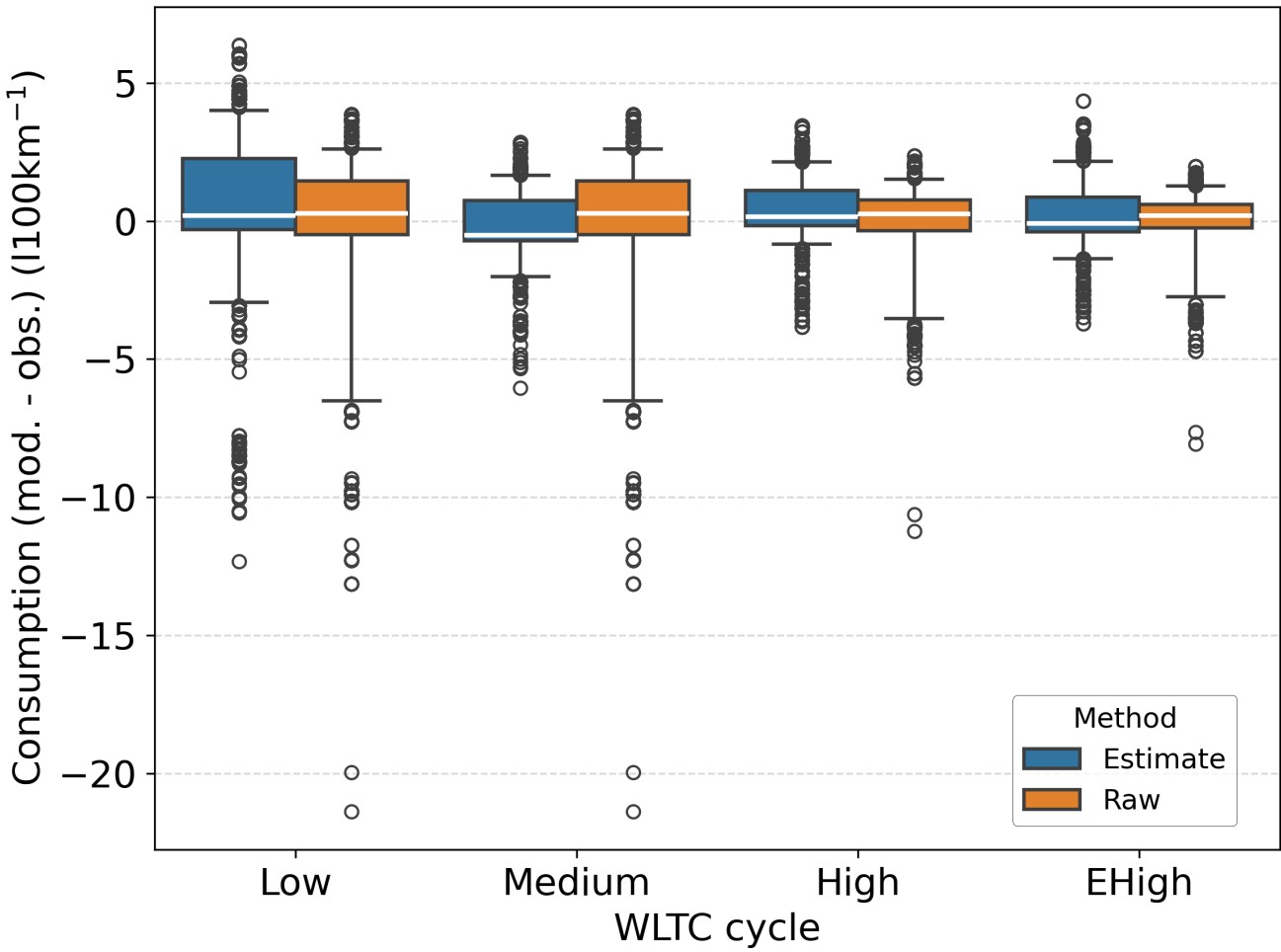

**Figure B2.** For each WLTC subcycle, boxplot of the difference between the fuel consumption modelled and the measurements for all the vehicles in the ADEME databank with the explicit method in blue and with a simple baseline which is the average fuel consumption for each WLTC subcycle in orange. The boxes extend from the first quartile ($Q_1$) to the third quartile ($Q_3$), with whiskers up to the farthest point lying within $1.5\times$ the interquartile range ($Q_3 - Q_1$).

Most fuel consumption estimates are within the range [-1, 2.5] (in litres per 100 km), as shown in Fig. B2. In this figure, outliers are composed of luxury and sports vehicles only, with a much higher fuel consumption. Since they are a very small part of the vehicle fleet, they are not representative of the behaviours of an average vehicle fleet. In addition, fuel consumption estimates are closer to the measured values as the WLTC subcycle increases. This can be explained by a lower fuel consumption variance between vehicle models as the mean speed increases. For instance, in the lower WLTC subcycles, the fuel consumption variance is larger. Thus, the estimates are less accurate. The baseline method performs better on average with lower variances in the low, high, and extra-high subcycles. However, the fuel consumption estimates are close enough to the real value to assume that the driver behaviour and engine efficiency estimates are satisfactory.

## Appendix C: Performances of the road surface temperatures simulated with the traffic impacts

TEB-CAR with the traffic impacts parameterised, and TEB are assessed with the road surface temperature (RST) observed in both directions. Common metrics are used to evaluate the performance of TEB-CAR and TEB, namely the mean squared error (MSE), the mean absolute error (MAE), the coefficient of determination ($R^2$) and the bias. A custom metrics is also used which is the mean time for a simulated variable to exceed a specific threshold in comparison with the observations named the average time threshold (ATT). Here, the ATT score calculate the average time the RST simulated reaches a value under 273.65 K in comparison with the observations.

The traffic impacts parameterised in TEB-CAR lead to higher performance compared to the TEB simulation with lower RMSE, MAE, and significantly higher $R^2$ for the entire simulation, as shown in Table C1. In particular, TEB-CAR corrects the larger temperature differences between the simulations and observations, as shown by the strong decrease in MSE and the reduced interquartile range in the Fig. C1 for both locations. There is a larger simulation error from the TEB model for the Nupuri location than for Palojärvi, but after simulation correction from the traffic-induced effects, they reach equivalent performance. Knowing that the traffic intensity is higher at Nupuri than at Palojärvi, it could mean that TEB-CAR reasonably reproduces the traffic-induced effect on the RST. The variance is reduced by about the same amount between the two locations, giving confidence in the quality of the modelling in two different scenarios with different traffic. Furthermore, the ATT score in Table C1 shows that TEB-CAR improves the accuracy to predict plausible dangerous conditions (RST < 273.65 K) by 1 K on average.

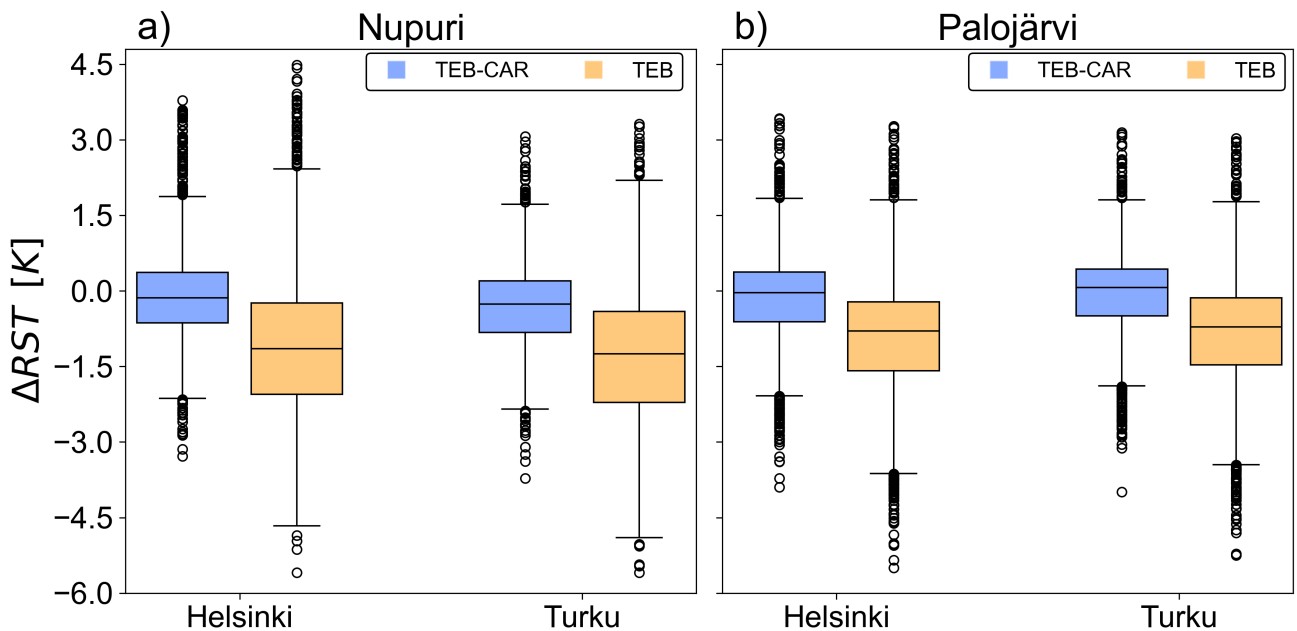

**Figure C1.** Boxplot of TEB and TEB-CAR simulations differences with the road surface temperature observations at Nupuri and Palojärvi location for the two directions, Helsinki and Tuku on joint Nupuri and Palojärvi period. The boxes extend from the first quartile ($Q_1$) to the third quartile ($Q_3$), with whiskers up to the farthest point lying within $1.5\times$ the interquartile range ($Q_3 - Q_1$).

| Exp. | TEB Tur. | TEB-CARt Tur. | TEB Hel. | TEB-CARh Hel. |
|---|---|---|---|---|
| MSE | 4.18 | 1.44 | 5.56 | 2.17 |
| MAE | 1.60 | 0.88 | 1.79 | 1.05 |
| Biais | -0.41 | -0.15 | -0.10 | 0.20 |
| R2 | 0.89 | 0.96 | 0.86 | 0.95 |
| ATT (RST < 273.65 K) | 3.01 | 1.97 | 3.22 | 2.25 |

**Table C1.** Comparison of the model TEB with the road surface temperature in the Turku direction (Tur.) and Helsinki direction (Hel.) against TEB-CAR simulation for Turku direction (TEB-CARt) and for Helsinki direction (TEB-CARh) during the entire simulation.

*Code and data availability.* TEB is embedded in the software SURFEX available from the CNRM open-source website: https://opensource.umr-cnrm.fr (CNRM, 2025) under the CeCILL Free Software License Agreement v1.0. The TEB-CAR module corresponding to the changes made in SURFEX V9.0, the raw data to construct the experiments, the preprocess script to prepare the simulations, the simulation configurations and results, and the scripts to reproduce the figures are available on the Zenodo platform (https://doi.org/10.5281/zenodo.17359513, Colas 2025).

*Author contributions.* GC built the methodology and conceptualization, conducted the formal analysis, validation, visualization and wrote the paper. VM, FB and LB planned and supervised the project, participated to the methodology ,the validation and proofread the paper.

*Competing interests.* The authors declare that they have no conflict of interest

*Disclaimer.*

*Acknowledgements.*

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
