# Peer review of "Traffic impact modelling in SURFEX-TEB V9.0 model for improved road surface temperature prediction"

_EGUsphere, 2025_

## Author Comment (AC1)

**Detailed response to referees comments**

We thank both reviewers for their comments on the study. The comments of both reviewers have induced changes in this new manuscript submission.

Thanks to the comment, we have made minor changes on the traffic impact parameterisations. First, we have modified the way we integrate the heat lost by the fuel consumption in the model. Rather than being considered as a ground fluxes, this flux is emitted directly as a component of the outgoing town sensible and latent heat flux. This calculation is more relevant for the tile approach in SURFEX. Second, for the analyses of the traffic impact on the road surface temperature, we consider its effect on the entire road lane width in Section 5.2. It allows a better understanding of the impact on average on the entire road surface. These changes modify marginally the results. However it does not changes the traffic impact interpretation.

Modifications have been made to clarify the methodology, the experimental set-up and the results. For the methodology, in the section 2 "Modelling strategy" new elements and clarifications have been made. For the experimental set-up, the paragraphs have been reordered and clarified. Finally, for the results the entire Section 5.2 named the "Analyses of the traffic impact parameterisations" has been completely rewritten.

In addition, on most graphics you will see changes, whether it is captions, label, or readability. One figure (Fig.8) has been completely changed in order to avoid misinterpretation and help the understanding for the reader.

We also have done a final sweep for grammar, spelling, and sentence clarity; thus you will see along the manuscript minor corrections.

We believe that thanks to these corrections, this work is now more understandable to the reader.

Best regards

**Major changes for the first reviewer (R.C 2)**

**R.C 2:** The current study focuses exclusively on winter. Could the authors elaborate on why summer was not included in the simulations? What differences in traffic-induced impacts would be expected between winter and summer conditions, particularly in terms of radiative processes, boundary layer development, and surface—atmosphere interactions?

**A.C:** We did not work on the summer, because the focus is on winter conditions and the key variable associated, the road surface temperature. In addition the road weather station data that has been given for this study correspond to the same period given by the openaccess data on the Zenodo from the FMI here. It runs from 19 October 2017 to 1 May 2018.

The spring conditions, subjected to more and more direct downward solar radiation as we approach the 1 May in the simulation, gives an idea of the impact for summer. The traffic-induced impacts would be relatively similar to the spring but at an higher extent. First, the heat lost from the combustion and the rolling friction are not weather dependent in this work, thus their extents will not change as their impact on the air and road surface temperature in summer. Second, the radiative impact of the vehicles is small compared to the other impacts. Thus, this impact is expected to stay small but to grow slightly. Finally, it will be interesting to see the effect on the wind-induced impact. This impact depends heavily on the difference between air and road surface temperatures. It leads to increased turbulence and heat exchange between air and road surface temperature. This impact is expected to grow as the temperature difference is would increase as well. In conclusion, the cumulative of these impacts may increase the overall cooling impact on the RST and the overall heating effect on the air temperature.

Changes in the manuscript: I484-I487 "Future work should evaluate the model subject to higher traffic intensities and summer conditions to assess the reliability of the traffic impacts parameterisation. The simulation period studied was relatively short, on winter and spring conditions only and only two road weather stations have been used to assess TEB-CAR improvements."

**R.C 2:** In the comparison with observations from the two road weather stations in southern Finland (Nupuri and Palojärvi), how were the atmospheric driving conditions specified in the model simulations? Were they based on in-situ measurements, reanalysis products, or another source? This information is important for assessing the reliability of the comparison.

**A.C:** The road weather stations measure the atmospheric variables needed to force the model (Specific humidity, air temperature and wind force and direction). In addition reanalysis of downward solar and infrared radiation from ERA5 are used to force the model.

We have modified the section 4 to increase the clarity of the experimental set-up. (1) The overall model configuration is now exclusively on the first paragraph. (2) Comments on why we choose Nupuri and Turku station are gathered in the second paragraph. (3) Everything related to the atmospheric forcings and the road weather stations specifications are gathered in paragraph four and five. We also have fixed the height symbol "h" in the Table 1.

**Changes in the manuscript:**

[revised manuscript text omitted]

**A.C:** The traffic-induced temperature difference asymmetry depends on the location and if it is modeled or observed. As shown in Fig.6, for Palojarvi, there is a traffic asymmetry with higher traffic intensity differences in the morning than in the afternoon. It leads on average to higher RST differences for both modeled and observed values.

For Nupuri, as shown in Fig.6, there is also an higher traffic intensity differences in the morning. It leads on the simulations to higher RST differences in the morning. In opposite, on the observed values, the RST differences are higher in the afternoon. On average the RST in the Turku direction is lower than on the Helsinki direction. This is true even when the traffic intensity differences are 0. The intercept of the regression equation on the observations gives -0.1846. These differences could be due to impacts from the surrounding elements or from sensor biases.

Changes in the manuscript: I376-I377 "The traffic intensity differences are lower in the afternoon than in the morning, leading to a lower observed  $\Delta RST_{dir}$ . These amplitudes are well reproduced by TEB-CAR."

**I391-I394** "At Nupuri location, the observed and simulated  $\Delta$ RSTdir distributions are shifted, the intercept of the regression equation for observed  $\Delta$ RSTdir is negative, and the observed  $\Delta$ RSTdir have a negative intercept. These features suggest that there is elements that produce this cold bias at Nupuri location that are not taken into account into TEB-CAR. It can also suggests that there is a sensor bias at this location since this effect is not found at Palojärvi location."

**R.C 2:** The phrase "During working days wekdays" appears to contain a typographical error. Please clarify or correct this expression.

**A.C:** This comment is relevant for the section "Analyses of the traffic impacts parameterisations". We have completely the section 5.2

---

## Author Comment (AC2)

**Detailed response to referees comments**

We thank both reviewers for their comments on the study. The comments of both reviewers have induced changes in this new manuscript submission.

Thanks to the comment, we have made minor changes on the traffic impact parameterisations. First, we have modified the way we integrate the heat lost by the fuel consumption in the model. Rather than being considered as a ground fluxes, this flux is emitted directly as a component of the outgoing town sensible and latent heat flux. This calculation is more relevant for the tile approach in SURFEX. Second, for the analyses of the traffic impact on the road surface temperature, we consider its effect on the entire road lane width in Section 5.2. It allows a better understanding of the impact on average on the entire road surface. These changes modify marginally the results. However it does not changes the traffic impact interpretation.

Modifications have been made to clarify the methodology, the experimental set-up and the results. For the methodology, in the section 2 "Modelling strategy" new elements and clarifications have been made. For the experimental set-up, the paragraphs have been reordered and clarified. Finally, for the results the entire Section 5.2 named the "Analyses of the traffic impact parameterisations" has been completely rewritten.

In addition, on most graphics you will see changes, whether it is captions, label, or readability. One figure (Fig.8) has been completely changed in order to avoid misinterpretation and help the understanding for the reader.

We also have done a final sweep for grammar, spelling, and sentence clarity; thus you will see along the manuscript minor corrections.

We believe that thanks to these corrections, this work is now more understandable to the reader.

Best regards

**Major changes for the first reviewer (R.C 1)**

**R.C 1 (Reviewer comments 1):** 2nd paragraph: Two more previous studies may also be interesting to the authors and provide insights. Chen et al. (2021) studied the 3D vehicle heat impact on the urban thermal environment in Hong Kong by modifying the WRF-SLUMC model, and later studied the impact of electric vehicles. The vehicle heat includes both spatial and temporal information. And consider the components of different vehicle types. Their studies also pointed out the seasonal variation of the vehicle heat impact 1.Chen, X., Yang, J.\*, Zhu, R., Wong, M. S., & Ren, C. (2021). Spatiotemporal impact of vehicle heat on urban thermal environment: A case study in Hong Kong. Building and Environment, 205, 108224. https://doi.org/10.1016/j.buildenv.2021.108224

2.Chen, X., & Yang, J.\* (2022). Potential benefit of electric vehicles in counteracting future urban warming: a case study of Hong Kong. Sustainable Cities and Society, 104200. https://doi.org/10.1016/j.scs.2022.104200

**A.C, (Author's comments 1):** We have added these citations to the 2nd paragraph and one sentence is added to be in agreement with these findings.

**Changes in the manuscript: I33-34** "However, the impact of traffic on the local climate is greater in winter, particularly at rush hours (Pigeon et al., 2007; Iamarino et al., 2011, Chen et al., 2021)."

I 31-32 "However, by adopting an electric-based vehicle fleet, the total anthropogenic heat released decreases proportionally Chen2022"

**R.C 1:** Modeling strategy, 1) How does the model consider different vehicle types providing different areas of shading? 2) And how does the model consider different vehicle types with different percentages of the total vehicle amount releasing different anthropogenic heat due to varied energy efficiency?

**A.C:** In this study, trucks, buses, and 2-wheels are omitted in this study. Thus, only passenger cars that are considered to build an estimate of traffic impacts.

The different areas of shading from the different passenger cars are taken into account indirectly through only one average length and width value as the impact on the solar and infrared net heat fluxes.

The same method is used to calculate engine efficiencies. A single equation is calculated, corresponding to the average engine efficiency of all different types of passenger cars. If the modeler wants to divide the vehicle types into several subclasses, it could be possible through the estimation of several engine efficiencies and with a weighted calculation. Below is an example for two types of vehicle, passenger cars and buses:

$$f_{traff} = (1-f_{sn})(p_{car}rac{\overline{w}}{w_{rd}}(rac{\overline{l}}{\overline{v}}\Phi) + p_{bus}rac{\overline{w_{bus}}}{w_{rd}}(rac{\overline{l_{bus}}}{\overline{v}}\Phi))$$

With  $p_{car} + p_{bus} = 1$

**Changes in the manuscript:**

**1351-354** "Finally, throughout the simulation period, the traffic counts show that more than 95% of the vehicles driven are passenger cars at both road weather stations (Colas, 2025). Therefore, to avoid complexity, only one vehicle type is considered for the estimation of the traffic parameters, with trucks, buses, and two-wheelers omitted. Estimates of the passenger cars engine efficiency and driver behaviors are made with the corresponding WLTC cycle and manufacturers' data (Colas, 2025)."

**1481-483** "It could be defined as a second vehicle type with its own set of estimated characteristics. The traffic impacts of the set of two average vehicle types could then be calculated as a simple arithmetic mean, weighted by a vehicle type ratio."

**R.C 1:** 3) How does the model consider whether there is a vehicle on the street or not at different times throughout the study period?

**A.C:** Traffic intensity is observed at the road weather station as an average number of vehicles per hour. These observations are given to the model and then converted to vehicles per second. Thus, at each model time step the number of vehicles is constant. We add two sentences for the reader.

**Changes in the manuscript: 193-95** "Traffic intensity is included in the model through average values of traffic counts and converted to vehicles per second. Therefore, traffic counts change at each atmospheric forcing time (each hour in this study)."

**R.C 1:** Methods: I suggest that authors list the surface energy balance before and after considering the traffic to demonstrate clearly how the traffic related process modifie the surface energy balance and then further changes the estimation of the surface temperature.

**A.C:** It will indeed help the reader to have a quick glance of the changes we have made. We have added the town energy balance equation before and after the traffic impact parameterisation as well as a quick explanation on how the traffic impact modifies the town energy balance.

**Changes in the manuscript: I113-127** "The urban energy balance without traffic impacts is written as:

$$Q^* + Q_f = Q_h + Q_e + \Delta Q_s + Q_m + \Delta Q_a$$

In the urban area,  $Q^*$  and Qf are the net radiative heat flux and the anthropic heat flux (without traffic), respectively. On the right-hand side of the equation, there are two turbulent fluxes Qh and Qe sensible and latent heat fluxes, respectively, and  $\Delta Qe$  the heat flux by conduction through the urban surfaces. Qm is the power exchanged by the melting and freezing of the water and finally  $\Delta Qm$  the horizontal advection, which is neglected in this study. All terms of the equation are expressed in watts per square metres.

Each traffic impact then modifies the energy balance within the canopy. First, three new source terms are added into this equation with the sensible QHengine and latent heat flux QEengine from the heat lost by the vehicle engine and the tyre-road friction heat flux Qr. Secondly, the turbulent fluxes Qh, Qe of the urban area are modified by the wind induced by the traffic Utraff. Indeed, traffic modifies the fluxes from the soil-atmosphere interaction. Finally, the solar and infrared net heat fluxes are modified by the vehicle body as depicted in Fig.~\ref{fig:Scheme} and the resulting energy from the vehicle balance is released as sensible heat in the air. The other terms are not formally modified, but react according to the new energy exchanges driven by the new and modified terms. The updated urban energy balance, function of the different traffic impacts gives:

$$Q^*(traff) + Q_f + Q_{f traff} = Q_h(U_{traff}) + Q_e(U_{traff}) + \Delta Q_s + Q_m + \Delta Q_a + Q_{Hengine} + Q_{Eengine} + Q_r + Q_{veh}$$
(2)

with Qf\_traff, the total source of energy coming from the vehicles distributed in the different right-hand terms of the equation.

**R.C 1:** Experimental set-up and model configurations: The estimation from ICCT needs more clarification; it is not clear to me with the current information.

**A.C:** We have refined the description of the traffic parameters extracted from ICCT and used in this study.

**Changes in the manuscript: I354-357** "In addition, missing input traffic parameters for the TEB-CAR simulations (average mass, length and height of the vehicles), are derived from the yearly passenger car statistics ICCT2023). In this study, the average vehicle body characteristics of the passenger cars sold in 2018 for the EU-28 are taken as input values and displayed in Table 1."

**R.C 1:** Results: I would suggest that authors list all simulation cases, including those with different physical processes again, to remind the reader.

**A.C:** To help the reader, several changes are made at the beginning of this section and on the experimental set-up section.

Changes in the manuscript: 1344-350 "In Sect. 5.2, an ablation setup is implemented. It means that for each road direction, 3 more simulations are launched, each with a traffic impact removed from the model. They are called rolling friction, radiative, and wind-induced. The heat released by combustion is not considered for this part because this flux is released on the upper vertical domain of the grid and so have no impact on the road surface temperature. By removing a traffic impact for each simulation, it is possible to investigate the relative impact of each traffic parameterisation on the simulated variables when compared with the reference simulation of TEB-CAR. To evaluate the impact of the traffic on the full road lane width, in this section the vehicle to road width ratio w/wrd is set to 0.5 in the traffic fraction occupation ftraff."

"The cumulative effect of the new set of traffic parameterisations in TEB (named TEB-CAR) results in marked impacts on the physical variables of the road. Each traffic parameterisation may have opposite or cumulative effects on the physical variables. In addition, each impact may change depending on atmospheric conditions, seasonality, and traffic intensity. Thus, in this section, the individual effect of each traffic impact in the model is studied using the Nupuri experiment throughout the entire simulation period. A total of eight more simulations at Nupuri are analysed here. Each simulation has a traffic impact removed from the model and is launched in both road directions."

**R.C 1:** Please maintain a consistent unit of temperature throughout the entire manuscript. *And also consider using different line styles*

A.C: All the °C units have been modified into Kelvin.

**R.C 1: Results:** Fig. 5a and b, are they Delta T or T, or the values subtracted from the observation values? Please make it clear.

**A.C:** In the article and in all the figures, we have modified the symbols for the road surface temperature. We now have 3 different symbols related to the road surface temperature. First: RST (K) for the road surface temperature. Second:  $\Delta RST_{dir}$  for the road surface temperature difference between the two road directions (it can be an observed value or a

simulated value). Third:  $\Delta RST_{mod}$  use only for the last Fig. 8. It represents the difference of temperature between TEB-CAR and TEB with (TEB-CAR minus TEB) In addition, we modified the captions for both figures.

Changes in the manuscript: Fig.5 "Comparison between the observed and simulated RST on a eight days subset period beginning a monday at Nupuri location. The subscripts "h" and "t" are for values on Helsinki and Turku directions respectively. From top to bottom panel: (a) observed and modelled road surface temperature difference between the two road directions  $\Delta RST_{dir}$  (Helsinki minus Turku), (b) road surface temperature, (c) number of vehicles per hour"

**Fig.6** "Road surface temperature differences  $\Delta RST_{dir}$  of both measured and simulated values between Helsinki and Turku directions (Helsinki minus Turku). Robust linear regression lines (RLM) are drawn between  $\Delta Traff$  and  $\Delta RST_{dir}$  for both observations and simulations. Panels (a) and (b) show the differences at the Palojärvi site on weekdays (a) and weekends (b). Panels (c) and (d) show the same information at the Nupuri site."

**R.C 1:** The lines in Fig. 7 are not very clear; consider increasing the height-to-width ratio of each sub-figure.

**A.C:** Thanks to this observation we have made the appropriate change for this figure by increasing the height of each subplot. We have also increased the readability of the new Fig. 8.

Changes in the manuscript:

**R.C 1:** Besides, for the same text in the two figures, do they indicate the same case? Please also clarify how to calculate the temperature difference ( $\Delta$ Ts). I suggest that the present values directly represent the impact, negative for cooling and positive for warming.

**A.C:** In agreement with this comment, to prevent misunderstanding and to emphasise on the traffic induced impacts, we have completely modified the figure. The new figure shows the road surface temperature differences between TEB-CAR and TEB (TEB-CAR minus TEB) and between the ablation experiment simulations and TEB on the Helsinki road direction. As suggested by the reviewer, a positive value shows a warming impact of the simulation without the wind-induced effect (in red). Thus, when compared to TEB-CAR, the wind-induced impact has an overall cooling effect.

**Changes in the manuscript:**

**R.C 1:** The figure legends in Figs. 7 and 8 are different; the same text is assigned a different color, which is confusing.

**A.C:** Indeed, it was supposed to be the same color for both figures, there is a mistake on the figure integrated in the manuscript. Thank you for pointing out this. The new Fig. 8 has the same color corresponding to the Fig.7 as you can see in the figure below and above.

**Changes in the manuscript:**

**R.C 1:** The results shown by Figures 7 and 8 are very interesting and important. However, the current description and discussion are not clear and comprehensive. I would encourage a more straightforward description and discussion of the mechanisms, for instance, including more discussion on the seasonal and diurnal variation.

The explanation and discussion of Fig. 7 are also not clear and robust enough. For example, the explanation of the less impact from the heat release is not robust and persuasive.

The whole description and discussion for Fig. 9 are missing.

**A.C:** It is true that this section is not detailed enough. We added some explanations. We have modified the entire section regarding the "Analyses of the traffic impacts parameterisations"

**Changes in the manuscript: Modified from I406 to I449**

[revised manuscript text omitted]